# unMORE: Unsupervised Multi-Object Segmentation via Center-Boundary Reasoning

Yafei Yang [1 2]   Zihui Zhang [1 2]   Bo Yang [1 2]

## Abstract

We study the challenging problem of unsupervised multi-object segmentation on single images. Existing methods, which rely on image reconstruction objectives to learn objectness or leverage pretrained image features to group similar pixels, often succeed only in segmenting simple synthetic objects or discovering a limited number of real-world objects. In this paper, we introduce **unMORE**, a novel two-stage pipeline designed to identify many complex objects in real-world images. The key to our approach involves explicitly learning three levels of carefully defined object-centric representations in the first stage. Subsequently, our multi-object reasoning module utilizes these learned object priors to discover multiple objects in the second stage. Notably, this reasoning module is entirely network-free and does not require human labels. Extensive experiments demonstrate that unMORE significantly outperforms all existing unsupervised methods across 6 real-world benchmark datasets, including the challenging COCO dataset, achieving state-of-the-art object segmentation results. Remarkably, our method excels in crowded images where all baselines collapse. Our code and data are available at https://github.com/vLAR-group/unMORE

## 1. Introduction

By age two, humans can learn around 300 object categories and recognize multiple objects in unseen scenarios (Frank et al., 2016). For example, after reading a book about the Animal Kingdom where each page illustrates a single creature, children can effortlessly recognize multiple similar animals at a glance when visiting a zoo, without needing additional

[1] Shenzhen Research Institute, The Hong Kong Polytechnic University; [2] vLAR Group, The Hong Kong Polytechnic University. . Correspondence to: Bo Yang <bo.yang@polyu.edu.hk>.

*Proceedings of the 42nd International Conference on Machine Learning*, Vancouver, Canada. PMLR 267, 2025. Copyright 2025 by the author(s).

teaching on site. Inspired by this efficient skill of perceiving objects and scenes, we aim to introduce a new framework to identify multiple objects from single images by learning object-centric representations, rather than relying on costly scene-level human annotations for supervision.

Existing works for unsupervised multi-object segmentation mainly consist of two categories: 1) Slot-based methods represented by SlotAtt (Locatello et al., 2020) and its variants (Sajjadi et al., 2022; Didolkar et al., 2024).

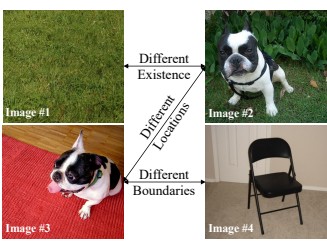

Figure 1: Object images.

They usually rely on an image reconstruction objective to drive the slot-structured bottlenecks to learn object representations. While achieving successful results on synthetic datasets (Karazija et al., 2021; Greff et al., 2022), they often fail to scale to complex real-world images (Yang & Yang, 2022; 2024). 2) Self-supervised feature distillation based methods such as TokenCut (Wang et al., 2022b), DINOSAUR (Seitzer et al., 2023), CutLER (Wang et al., 2023a), and CuVLER (Arica et al., 2024). Thanks to the strong object localization hints emerging from self-supervised pretrained features such as DINO/v2 (Caron et al., 2021; Oquab et al., 2023), these methods explore this property to discover multiple objects via feature reconstruction or pseudo mask creation for supervision. Despite obtaining very promising segmentation results on real-world datasets such as COCO (Lin et al., 2014), they still fail to discover a satisfactory number of objects. Primarily, this is because the simple feature reconstruction or pseudo mask creation for supervision tends to distill or define rather weak objectness followed by ineffective object search, resulting in only a few objects correctly discovered. In fact, unsupervised multi-object segmentation of a single image is hard and not straightforward, as it involves two critical issues: 1) the definition of *what objects are (i.e., objectness)* is unclear, 2) there is a lack of an effective way to discover *those objects* in unseen scenes.

In this paper, to tackle these issues, we propose a two-stage pipeline consisting of an object-centric representation learn-

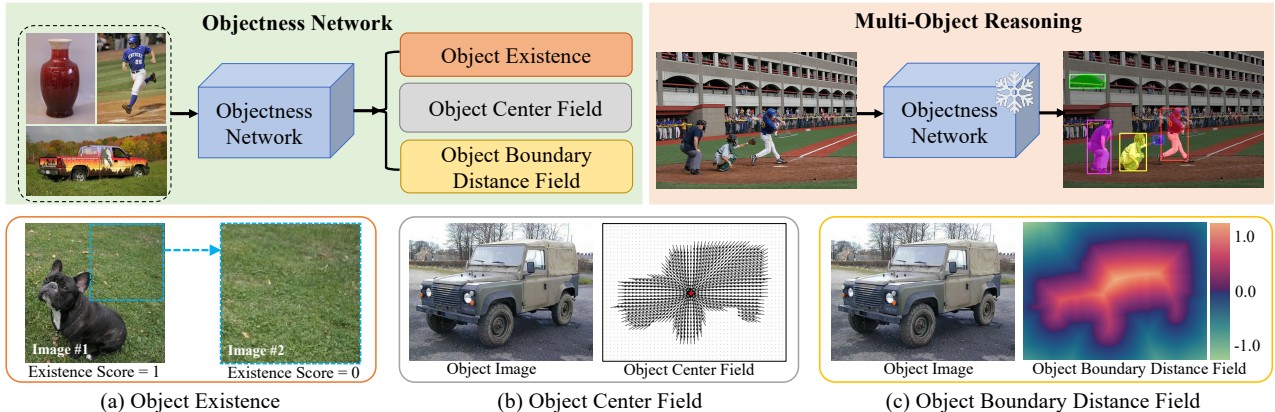

(a) Object Existence     (b) Object Center Field     (c) Object Boundary Distance Field

Figure 2: The upper blocks illustrate our framework. The lower blocks show three levels of object-centric representations.

ing stage followed by an effective multi-object reasoning stage, akin to a human's innate skill of perceiving objects and scenes. As illustrated in the upper left block of Figure 2, in the first stage, we aim to train an **objectness network** to learn our explicitly defined object-centric representations from monolithic object images such as those in ImageNet. In the second stage, as illustrated in the right block of Figure 2, we introduce a **multi-object reasoning module** to automatically discover individual objects in single images just by querying our pretrained and frozen objectness network, instead of requiring human annotations for supervision.

Regarding the **objectness network**, our key insight is that, given an input image or patch, it should be able to answer three essential questions: 1) is there an object inside (*i.e.*, *object existence*)? 2) if so, where is it (*i.e.*, *object location/center*)? and 3) what is the object shape (*i.e.*, *object boundary*)? Essentially, training such an objectness network would be analogous to the learning process of infants forming concepts of objects in their minds. As illustrated in Figure 1, we can see that there is no salient object in image #1, but images #2/#3 contain similar dogs at different locations, whereas image #4 has another object with different shape boundaries. By training on such images, our objectness network aims to explicitly capture these top-down (existence/location) and bottom-up (boundary) object-centric representations. To achieve this goal, we introduce three corresponding levels of objectness to learn in parallel: 1) a binary object existence score, 2) an object center field, and 3) an object boundary distance field, as shown in Figure 2.

With respect to the **multi-object reasoning module**, we aim to discover as many individual objects as possible in scene-level images. Our insight is that, given a multi-object image, if a cropped patch has a single valid object inside, its three levels of objectness representations must satisfy a certain threshold when querying against our pretrained objectness network. Otherwise, that patch should be discarded or its position and size should be effectively updated until a valid object is included inside. To this end, we introduce a center-

boundary-aware reasoning algorithm to iteratively regress accurate multi-object bounding boxes and masks according to the learned three levels of object-centric representations from our pretrained objectness network. Notably, the multi-object reasoning is completely network-free and requires no human labels for supervision.

Our framework, named **unMORE**, learns object-centric representations through the objectness network, enabling **un**supervised **m**ulti-**o**bject **re**asoning on single images. Our contributions are:

- We introduce a new pipeline comprising object-centric learning and multi-object reasoning, and propose three levels of explicit object-centric representations, including object existence, object center field, and object boundary distance field learned by an objectness network.

- We design a center-boundary aware reasoning algorithm to iteratively discover multiple objects in single images. The algorithm is network-free and human-label-free.

- We demonstrate superior object segmentation results and clearly surpass state-of-the-art unsupervised methods on 6 benchmark datasets including the challenging COCO.

## 2. Related Work

**Object-centric Learning without Pretrained Features**: Object-centric learning involves the unsupervised discovery of multiple objects in a scene. A plethora of methods have been proposed in the past years (Yuan et al., 2023). They primarily rely on an image reconstruction objective to learn objectness from scratch without needing any human labels or pretrained image features. Early models aim to learn object factors such as size, position, and appearance from raw images by training (variational) autoencoders (AE/VAE) (Kingma & Welling, 2014), including AIR (Eslami et al., 2016), SPACE (Lin et al., 2020), and others (Greff et al., 2016; 2017; Crawford & Pineau, 2019; Burgess et al., 2019; Greff et al., 2019). Recently, with the success of slot-based methods (Locatello et al., 2020; Engelcke et al.,

2020), most succeeding works (Engelcke et al., 2021; Sajjadi et al., 2022; Löwe et al., 2022; Biza et al., 2023; Löwe et al., 2023; Foo et al., 2023; Brady et al., 2023; Jia et al., 2023; Stanić et al., 2023; Lachapelle et al., 2023; Kirilenko et al., 2024; Gopalakrishnan et al., 2024; Wiedemer et al., 2024; Didolkar et al., 2024; Mansouri et al., 2024; Kori et al., 2024a;b; Jung et al., 2024; Fan et al., 2024) extend the slot structure from various aspects to improve the object segmentation performance. Although achieving excellent results, they often fail to scale to complex real-world images as investigated in (Yang & Yang, 2022; 2024). To overcome this limitation, a line of works (Weis et al., 2021) use additional information such as motion and depth to identify objects. Unfortunately, this precludes learning on most real-world images, which do not have motion or depth information.

**Object-centric Learning with Pretrained Features**: Very recently, with the advancement of self-supervised learning techniques, strong object semantic and localization hints emerge from these features, like DINO/v2 (Caron et al., 2021; Oquab et al., 2023) pretrained on ImageNet (Deng et al., 2009) without any annotation. An increasing number of methods leverage such features for unsupervised salient/single object detection (Voynov et al., 2021; Shin et al., 2022a; Tian et al., 2024), or multi-object segmentation (Siméoni et al., 2024), or video object segmentation (Aydemir et al., 2023; Zadaianchuk et al., 2024). Representative works include the early LOST (Siméoni et al., 2021), ODIN (Hénaff et al., 2022), TokenCut (Wang et al., 2022b), and the recent DINOSAUR (Seitzer et al., 2023), CutLER (Wang et al., 2023a), and UnSAM (Wang et al., 2024). These methods and their variants (Wang et al., 2022a; Singh et al., 2022; Ishtiak et al., 2023; Wang et al., 2023c;b; Niu et al., 2024; Zhang et al., 2024) achieve very promising object segmentation results on challenging real-world datasets, demonstrating the value of pretrained features. However, they still fail to discover a satisfactory number of objects, and the estimated object bounding boxes and masks often suffer from under-segmentation issues. Essentially, this is because these methods tend to simply group pixels with similar features (obtained from pretrained models) as a single object, lacking the ability to discern boundaries between objects. As a consequence, for example, they usually group two chairs nearby into just one object. By contrast, our introduced three level object-centric representations are designed to jointly retain unique and explicit objectness features for each pixel, *i.e.*, how far away to the object boundary and in what direction to the object center.

**Object-centric Representations**: To represent objects for downstream tasks such as detection, segmentation, matching, and reconstruction, various properties can be used, including object center/centroid, object binary mask (Cai & Vasconcelos, 2018; Cheng et al., 2022), and object boundary (Park et al., 2019). For example, prior works

(Gall & Lempitsky, 2009; Gall et al., 2011; Qi et al., 2019; Ahn et al., 2019) learn to transform pixels/points to object centroids for better segmentation, and the works (Thanh Nguyen, 2014; Ma et al., 2010) use object boundaries as the template for shape matching. However, these works are primarily designed for fully supervised tasks, whereas we focus on learning object-centric representations for unsupervised multi-object segmentation. In particular, our carefully designed three-level object-centric representations aim to jointly describe objects in a nuanced manner, and our unique multi-object reasoning module can make full use of the learned object-centric representations to tackle under- and over-segmentation issues.

## 3. unMORE

### 3.1. Preliminary

Our objectness network aims to learn three levels of object-centric representations from the large-scale ImageNet dataset. Thanks to the advanced self-supervised learning techniques, which give us semantic and location information of objects in pretrained models, we opt to use pretrained features to extract object regions on ImageNet to bootstrap our objectness network.

In particular, we exactly follow the VoteCut method proposed in CuVLER (Arica et al., 2024) to obtain a single object mask (binary) on each image of ImageNet. First, each image of ImageNet is fed into the self-supervised pretrained DINO/v2, obtaining patch features. Second, an affinity matrix is constructed based on the similarity of patch features, followed by Normalized Cut (Shi & Malik, 2000) to obtain multiple object masks. Third, the most salient mask of each image is selected as the rough foreground object. For more details, refer to CuVLER. These rough masks will be used to learn our object-centric representations in Section 3.2.

### 3.2. Objectness Network

With single object images and the prepared (rough) masks on ImageNet (the object image denoted as $I \in \mathcal{R}^{H \times W \times 3}$, object mask as $M \in \mathcal{R}^{H \times W \times 1}$), the key to train our objectness network is the definitions of three levels of object-centric representations, which are elaborated as follows.

**Object Existence Score**: For an image $I$, its object existence score $f^e$ is simply defined as 1 (positive sample) if it contains a valid object, *i.e.*, $sum(M) >= 1$, and 0 otherwise (negative sample). In the preliminary stage of processing ImageNet, since every image contains a valid object, we then create a twin negative sample by cropping the largest rectangle on background pixels excluding the tightest object bounding box. As illustrated in Figure 2 (a), image #1 is an original sample from ImageNet, whereas image #2 is a twin negative sample created by us.

**Object Center Field**: For an image $\boldsymbol{I}$ with a valid object mask $\boldsymbol{M}$ inside, its object center field $\boldsymbol{f}^c$ is designed to indicate the position/center of the object, *i.e.*, the tightest object bounding box center. As illustrated in Figure 2(b), each pixel within the object mask is assigned with a unit vector pointing to the object center $[C_h, C_w]$, and pixels outside the mask are assigned zero vectors. Formally, the center field value at the $(h, w)^{th}$ pixel, denoted as $\boldsymbol{f}^c_{(h,w)}$, is defined as follows, where $\boldsymbol{f}^c \in \mathcal{R}^{H \times W \times 2}$. Basically, this center field aims to capture the relative position of an object with respect to the pixels of an image.

$$\boldsymbol{f}^c_{(h,w)} = \begin{cases} \frac{[h,w]-[C_h,C_w]}{\|[h,w]-[C_h,C_w]\|}, & \text{if } \boldsymbol{M}_{(h,w)} = 1 \\ [0,0], & \text{otherwise} \end{cases} \quad (1)$$

We notice that prior works use Hough Transform to transform pixels/points to object centroids for 2D/3D object detection (Gall et al., 2011; Qi et al., 2019), which requires learning both directions and distances to object centers. However, our object center field is just defined as unit directions pointing to object centers, as we only need to learn such directions to identify multi-center proposals instead of recovering object masks as detailed in Step #2 of Sec 3.3.

**Object Boundary Distance Field**: For the same image $\boldsymbol{I}$ and its object mask $\boldsymbol{M}$, this boundary distance field $\boldsymbol{f}^b$ is designed to indicate the shortest distance from each pixel to the object boundary. To discriminate whether a pixel is inside or outside of an object, we first compute the simple signed distance field, where the distance values inside the object mask are assigned to be positive, those outside are negative, and boundary pixels are zeros. This signed distance field is denoted as $\boldsymbol{S} \in \mathcal{R}^{H \times W \times 1}$ for the whole image, and its value at the $(h, w)^{th}$ pixel $S_{(h,w)}$ is calculated as follows:

$$S_{(h,w)} = \begin{cases} \|[h,w]-[\bar{h},\bar{w}]\|, & \text{if } \boldsymbol{M}_{(h,w)} = 1 \\ -\|[h,w]-[\bar{h},\bar{w}]\|, & \text{otherwise} \end{cases} \quad (2)$$

where the location $(\bar{h}, \bar{w})$ is the nearest pixel position on the object boundary corresponding to the pixel $(h, w)$. Detailed steps of calculation are in Appendix A.1. These signed distance values are measured by the number of pixels and could vary significantly across images with differently-sized objects. Notably, the maximum signed distance value within an object mask $\boldsymbol{M}$, assuming it appears at the $(\hat{h}, \hat{w})^{th}$ pixel location, *i.e.*, $S_{(\hat{h},\hat{w})} = max(\boldsymbol{S} * \boldsymbol{M})$, indicates the object size. The higher $S_{(\hat{h},\hat{w})}$, the more likely the object is larger or its innermost pixel is further away from the boundary.

To stabilize the training process, we opt to normalize signed distance values as our object boundary distances. Notably, signed distances for foreground and background are normalized separately. For the $(h, w)^{th}$ pixel, our object boundary

distance field, denoted as $\boldsymbol{f}^b_{(h,w)}$, is defined as follows:

$$\boldsymbol{f}^b_{(h,w)} = \begin{cases} \frac{S_{(h,w)}}{max(\boldsymbol{S}*\boldsymbol{M})}, & \text{if } \boldsymbol{M}_{(h,w)} = 1 \\ \frac{S_{(h,w)}}{|min(\boldsymbol{S}*(\mathbf{1}-\boldsymbol{M}))|}, & \text{otherwise} \end{cases} \quad (3)$$

where * represents element-wise multiplication and $\boldsymbol{f}^b \in \mathcal{R}^{H \times W \times 1}$. Figure 2(c) shows an example of an object image and its final boundary distance field. Our above definition of the boundary distance field has a nice property that the maximum signed distance value $S_{(\hat{h},\hat{w})}$ can be easily recovered based on the norm of the gradient of $\boldsymbol{f}^b$ at any pixel inside of object as follows. This property is crucial to quickly search for object boundaries at the stage of multi-object reasoning as discussed in Section 3.3.

$$S_{(\hat{h},\hat{w})} = 1 / \|[\frac{\partial \boldsymbol{f}^b_{(h,w)}}{\partial h}, \frac{\partial \boldsymbol{f}^b_{(h,w)}}{\partial w}]\|, \quad \text{if } \boldsymbol{f}^b_{(h,w)} > 0 \quad (4)$$

Notably, the concept of the boundary distance field (Park et al., 2019; Xie et al., 2022) is successfully used for shape reconstruction. Here, we demonstrate its effectiveness for object discovery.

Overall, for all original images of ImageNet, three levels of object-centric representations are clearly defined based on the generated rough object masks in Section 3.1. We also create twin negative images with zero existence scores.

**Objectness Network Architecture and Training**: Having the defined representations on images, we simply choose two commonly-used existing networks in parallel as our objectness network, particularly, using ResNet50 (He et al., 2016) as a binary classifier to predict *object existence scores* $\tilde{f}^e$, and using DPT-large (Ranftl et al., 2021) followed by two CNN-based heads to predict *object center field* $\tilde{\boldsymbol{f}}^c$ and *object boundary distance field* $\tilde{\boldsymbol{f}}^b$ respectively. To train the whole model, the cross-entropy loss is applied for learning existence scores, L2 loss for the center field, and L1 loss for the boundary distance field. Our total loss is defined as follows and more details are provided in Appendix A.2.

$$\ell = CE(\tilde{f}^e, f^e) + \ell_2(\tilde{\boldsymbol{f}}^c, \boldsymbol{f}^c) + \ell_1(\tilde{\boldsymbol{f}}^b, \boldsymbol{f}^b) \quad (5)$$

### 3.3. Multi-Object Reasoning Module

With the objectness network well-trained on ImageNet, our ultimate goal is to identify as many objects as possible on complex scene images without needing human labels for supervision. Given a single scene image, a naïve solution is to endlessly crop many patches with different resolutions at different locations, and then feed them into our pretrained objectness network to verify each patch's objectness. However, this is inefficient and infeasible in practice. To this end, we introduce a network-free multi-object reasoning module consisting of the following steps.

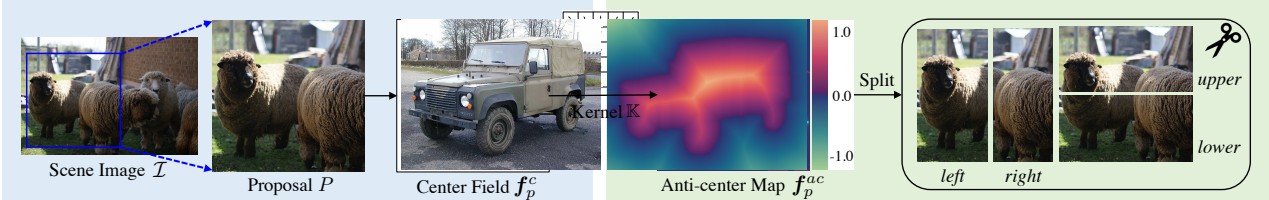

Figure 3: An illustration of kernel-based operation for multi-center detection and proposal splitting.

**Step #0 - Initial Object Proposal Generation**: Given a scene image $\mathcal{I} \in \mathcal{R}^{M \times N \times 3}$, we randomly and uniformly initialize a total of $T$ bounding box proposals by selecting a set of anchor pixels on the entire image. At each anchor pixel, multiple sizes and aspect ratios are chosen to create initial bounding boxes. More details are provided in Appendix A.3. For each proposal $P$, its top-left and bottom-right corner positions in the original scene image will always be tracked and denoted as $[P^{u_1}, P^{v_1}, P^{u_2}, P^{v_2}]$. We also linearly scale up or down all proposals to have the same resolution of $128 \times 128$ to feed into our objectness network subsequently.

**Step #1 - Existence Checking**: For each bounding box proposal $P$, we feed the corresponding image patch (cropped from $\mathcal{I}$) into our pretrained and frozen objectness network, querying its existence score $f_p^e$. The proposal will be discarded if $f_p^e$ is smaller than a threshold $\tau^e$. The higher $\tau^e$ is, the more aggressive it is to ignore potential objects.

**Step #2 - Center Reasoning**: For the proposal $P$ with a high enough object existence score, we then query its center field $f_p^c$ from our objectness network. This step #2 aims to evaluate whether $f_p^c$ has only one center or $\geq 2$ centers. If there is just one center, the non-zero center field vectors of $f_p^c$ are likely pointing to a common position. Otherwise, those vectors are likely pointing to multi-positions. In the latter case, the proposal $P$ needs to be safely split into subproposals at pixels whose center field vectors are facing opposite directions. Thanks to this nice property, we propose the following simple kernel-based operation for multi-center detection and proposal splitting.

As shown in the left block of Figure 3, given the center field $f_p^c \in \mathcal{R}^{128 \times 128 \times 2}$ of a proposal $P$, we predefine a kernel $\mathbb{K} \in \mathbf{R}^{5 \times 5 \times 2}$ where each of the $(5 \times 5)$ vectors has a unit length and points outward against the kernel center. Details of kernel values are in Appendix A.3. By applying this kernel on top of $f_p^c$ with a stride of $1 \times 1$ and zero-paddings, we obtain an anti-center map, denoted as $f_p^{ac} \in \mathcal{R}^{128 \times 128 \times 1}$. The higher the anti-center value at a specific pixel, the more likely that pixel is in between multiple crowded objects. Otherwise, that pixel is more likely to be near an object center or belongs to the background. Clearly, the former case is more likely to incur under-segmentation.

For this anti-center map $f_p^{ac}$ of the proposal $P$, 1) if its highest value among all pixels is greater than a threshold $\tau^c$, this proposal $P$ is likely to have $\geq 2$ crowded objects and will be split at the corresponding pixel location with the highest value. As shown in the right block of Figure 3, we safely split the proposal $P$ into 4 subproposals at the highest anti-center value (yellow star): $\{left, right, upper, lower\}$ halves. Each subproposal is regarded as a brand-new one and will be evaluated from **Step #1** again. With this design, the particularly challenging under-segmentation issue often incurred by multiple crowded objects can be resolved.

2) If the highest value of $f_p^{ac}$ is smaller than the threshold $\tau^c$, the proposal $P$ is likely to have just one object, or multiple objects but they are far away from each other, *i.e.*, more than 5 pixels apart. In this regard, we simply adopt the connected-component method used in CuVLER (Arica et al., 2024) to split the proposal $P$ into subproposals. Specifically, for its center field $f_p^c$, all pixels that are spatially connected and have non-zero unit vectors are grouped into one subproposal. Each subproposal is regarded as a brand-new one and will be evaluated from **Step #1** again.

**Step #3 - Boundary Reasoning**: At this step, the proposal $P$ is likely to have a single object, and we query its boundary distance field $f_p^b$ from our objectness network. The ultimate goal of this step is to correctly update this proposal's location and size, *i.e.*, the two corner positions $[P^{u_1}, P^{v_1}, P^{u_2}, P^{v_2}]$ in its original scene image $\mathcal{I}$, such that the proposal can converge to a tight bounding box of the object inside. Recall that, in Equations 3&4, our definition of the boundary distance field and its gradient have a crucial property. Particularly, the value at a specific pixel of the boundary distance field $f_p^b$ indicates how far it is away from the nearest object's boundaries. This means that we can directly use $f_p^b$ to help update the two corner positions.

Intuitively, if the proposal $P$ has an incomplete object, its borders need to expand. If it has many background pixels, its borders need to contract. With this insight, we only need to focus on boundary distance values of the four borders of $f_p^b$ to decide the margins to expand or contract. To this end, we introduce the following border-based reasoning algorithm to update $[P^{u_1}, P^{v_1}, P^{u_2}, P^{v_2}]$.

As illustrated in Figure 4, for the boundary distance field $f_p^b \in \mathcal{R}^{128 \times 128 \times 1}$ of a proposal $P$, we first collect values at four borders {*topmost row, leftmost column, bottommost row, rightmost column*} highlighted by red dotted lines, denoted by four vectors: $\{f_{p_t}^b, f_{p_l}^b, f_{p_b}^b, f_{p_r}^b\} \in \mathcal{R}^{128}$. Each of the

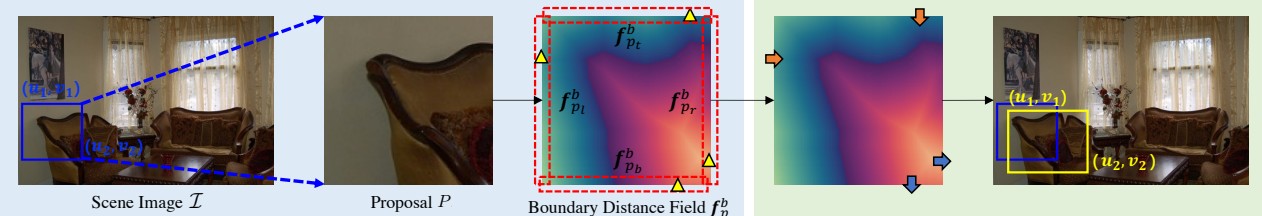

Figure 4: An illustration of border-based reasoning algorithm to update proposals.

four borders of proposal $P$ is updated as follows:

$$P^{u_1} \leftarrow P^{u_1} - \frac{max(\boldsymbol{f}_{p_t}^b)}{\|[\frac{\partial \boldsymbol{f}_{p_t}^b}{\partial u}, \frac{\partial \boldsymbol{f}_{p_t}^b}{\partial v}]\|}, (u,v) = argmax \boldsymbol{f}_{p_t}^b \quad (6)$$

$$P^{v_1} \leftarrow P^{v_1} - \frac{max(\boldsymbol{f}_{p_l}^b)}{\|[\frac{\partial \boldsymbol{f}_{p_l}^b}{\partial u}, \frac{\partial \boldsymbol{f}_{p_l}^b}{\partial v}]\|}, (u,v) = argmax \boldsymbol{f}_{p_l}^b$$

$$P^{u_2} \leftarrow P^{u_2} + \frac{max(\boldsymbol{f}_{p_b}^b)}{\|[\frac{\partial \boldsymbol{f}_{p_b}^b}{\partial u}, \frac{\partial \boldsymbol{f}_{p_b}^b}{\partial v}]\|}, (u,v) = argmax \boldsymbol{f}_{p_b}^b$$

$$P^{v_2} \leftarrow P^{v_2} + \frac{max(\boldsymbol{f}_{p_r}^b)}{\|[\frac{\partial \boldsymbol{f}_{p_r}^b}{\partial u}, \frac{\partial \boldsymbol{f}_{p_r}^b}{\partial v}]\|}, (u,v) = argmax \boldsymbol{f}_{p_r}^b$$

Because $\{max(\boldsymbol{f}_{p_t}^b), max(\boldsymbol{f}_{p_l}^b), max(\boldsymbol{f}_{p_b}^b), max(\boldsymbol{f}_{p_r}^b)\}$ could be positive or negative, this results in the four borders of the proposal $P$ expanding or contracting by themselves. As shown in the rightmost block of Figure 4, the proposal $P$ is updated from the blue rectangle to the yellow one whose bottom and right borders expand to include more object parts because their maximum boundary distance values are positive, whereas its top and left borders contract to exclude more background pixels because their maximum boundary distance values are negative. As boundary distance values are physically meaningful, each expansion step will not go far outside of the tightest bounding box and each contraction step will not go deep into the tightest bounding box.

Among the total four steps, the center-boundary-aware reasoning **Steps #2/#3** are crucial and complementary to tackle the core under-/over-segmentation issues. Once the two corners of a proposal $P$ are updated, we will feed the updated proposal into **Step #3** until the corners converge to stable values. During this iterative updating stage, we empirically find that it is more efficient to take a slightly larger step size for expansion, and a smaller step size for contraction. More details are in Appendix A.3. The efficiency of our direct iterative updating is also investigated in Appendix A.18.

Once the size and location of a proposal $P$ converge, a valid object is discovered. After all proposals are processed in parallel through **Steps #1/#2/#3**, we collect all bounding boxes and apply the standard NMS to filter out duplicate detections. For each final bounding box, we obtain its object mask by taking the union of positive values within its boundary distance field and non-zero vectors within its center field. We also compute a confidence score for each object based on its object existence score, center field, and boundary distance field. More details are in Appendix A.4.

Overall, with the pretrained objectness network in Section 3.2, and the network-free multi-object reasoning module in Section 3.3, our pipeline can discover multiple objects in single scene images without training an additional detector. This pipeline is named as **unMORE**$_{disc}$ in experiments.

**Optionally Training a Detector**: As shown in CutLER (Wang et al., 2023a) and CuVLER (Arica et al., 2024), the discovered objects from scene images can be used as pseudo labels to train a separate class agnostic detector (CAD) from scratch. We select and weight each discovered object based on its confidence score. Intuitively, the selected objects should have high object existence scores, homogeneous center fields and boundary fields. More details about the pseudo label selection are provided in Appendix A.5.

Following CuVLER, we also train a class agnostic detector using the same network architecture and training strategy based on our own pseudo labels from scratch. Our trained detector is named as **unMORE** in experiments.

## 4. Experiments

**Datasets:** Evaluation of existing unsupervised multi-object segmentation methods is primarily conducted on the challenging COCO validation set (Lin et al., 2014). However, we empirically find that a large number of objects are actually not annotated in validation set. This may not be an issue for evaluating fully-supervised methods in literature, but likely gives an inaccurate evaluation of unsupervised object discovery. To this end, we further manually augment object annotations of COCO validation set by labelling additional 197 object categories. It is denoted as **COCO\*** validation set and will be released to the community. Details of the additional annotations are in Appendix A.16. We also evaluate on datasets of **COCO20K** (Lin et al., 2014), **LVIS** (Gupta et al., 2019), **VOC** (Everingham et al., 2010), **KITTI** (Geiger et al., 2012), **Object365** (Shao et al., 2019), and **OpenImages** (Kuznetsova et al., 2020).

**Baselines:** For an extensive comparison on COCO\* validation set, we include the following three groups of methods.

*Group 1 - Direct Object Discovery w/o Learnable Modules.* The following methods directly discover objects from COCO\* val set, without involving any trainable modules.

- **FreeMask**: proposed in FreeSOLO (Wang et al., 2022a) to discover multi-objects based on DenseCL features.

Table 1: Quantitative results on COCO* val set. "# of pred obj." refers to the average number of predicted objects per image.

| | | Trainable Module | AP$_{50}^{box}$ | AP$_{75}^{box}$ | AP$^{box}$ | AR$_{100}^{box}$ | AR$^{box}$ | AP$_{50}^{mask}$ | AP$_{75}^{mask}$ | AP$^{mask}$ | AR$_{100}^{mask}$ | AR$^{mask}$ | # of pred obj. |
|---|---|---|---|---|---|---|---|---|---|---|---|---|---|
| Direct Object Discovery | w/o Learnable Modules | FreeMask | - | 3.7 | 0.6 | 1.3 | 4.6 | 4.6 | 3.1 | 0.3 | 0.9 | 3.5 | 3.5 | 3.7 |
| | | MaskCut (K=3) | - | 6.0 | 2.4 | 2.9 | 6.7 | 6.7 | 5.1 | 1.8 | 2.3 | 5.8 | 5.8 | 1.8 |
| | | MaskCut (K=10) | - | 6.2 | 2.6 | 2.9 | 7.2 | 7.2 | 5.3 | 2.0 | 2.3 | 6.2 | 6.2 | 2.1 |
| | | VoteCut | - | 10.8 | 4.9 | 5.5 | 11.3 | 11.3 | 9.5 | 4.0 | 4.6 | 9.8 | 9.8 | 8.9 |
| | w/ Learnable Modules | DINOSAUR | Recon. SlotAtt | 2.0 | 0.2 | 0.6 | 4.8 | 4.8 | 1.1 | 0.1 | 0.3 | 2.9 | 2.9 | 7.0 |
| | | FOUND | Seg. Head | 4.4 | 1.8 | 2.1 | 3.6 | 3.6 | 3.3 | 1.3 | 1.5 | 3.0 | 3.0 | 1.0 |
| | | **unMORE$_{disc}$(Ours)** | Obj. Net | 19.1 | 9.0 | 10.1 | 19.6 | 19.6 | 17.8 | 8.7 | 9.5 | 18.9 | 18.9 | 8.2 |
| Training Detectors | - | UnSAM | Detector x 4 | 10.2 | 6.3 | 6.4 | 36.1 | **50.1** | 10.2 | 6.2 | 6.3 | 34.1 | **46.1** | 332.2 |
| | | CutLER | Detector x 3 | 26.0 | 14.2 | 14.7 | 37.9 | 37.9 | 22.7 | 11.2 | 11.8 | 32.7 | 32.7 | 100.0 |
| | | CuVLER | Detector x 2 | 28.0 | 14.8 | 15.5 | 37.8 | 37.8 | 24.4 | 11.7 | 12.6 | 32.1 | 32.1 | 99.7 |
| | | **unMORE(Ours)** | **Obj. Network + Detector x 1** | **32.6** | **17.2** | **18.0** | **40.9** | 40.9 | **29.6** | **14.4** | **15.5** | **36.5** | 36.5 | 100.0 |

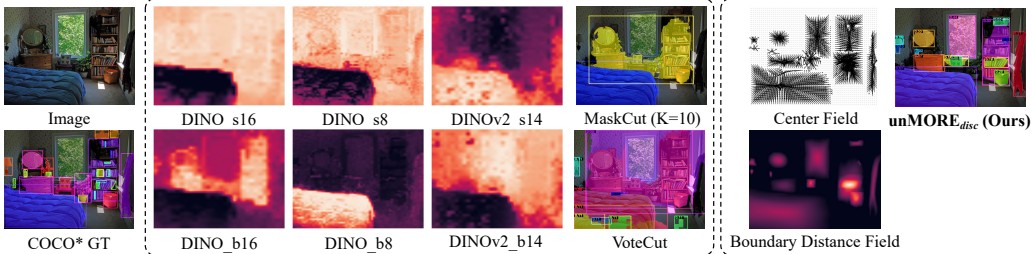

Figure 5: Results on COCO* validation set. For MaskCut and VoteCut, the eigenvectors of the second smallest eigenvalue for their used DINO/v2 features are visualized. For unMORE$_{disc}$, center and boundary object representations are visualized.

- **MaskCut**: proposed in CutLER (Wang et al., 2023a) to discover multi-objects based on DINO features. The number of cut $K$ is set as both 3 and 10 in its favor.
- **VoteCut**: proposed in CuVLER (Arica et al., 2024) to discover multi-objects based on DINO/v2 features.

*Group 2 - Direct Object Discovery w/ Learnable Modules.* The following methods use learnable modules to aid object discovery, but without training any multi-object detector.

- **DINOSAUR** (Seitzer et al., 2023): It discovers multi-objects by learning to reconstruct DINO features.
- **FOUND** (Siméoni et al., 2023): This is a salient object detection method.
- **unMORE$_{disc}$** (Ours): We discover multi-objects by network-free reasoning through our objectness network.

*Group 3 - Object Segmentation by Training Additional Multi-object Detectors.* The following methods discover objects by training additional detectors. We adopt a diverse range of settings for each method and report the highest scores from their best setting. A full list of all settings and results are in Appendix A.9. Note that, all final evaluation is conducted on COCO* val set which is completely held out.

- **CutLER**: Its best setting is to train detectors on pseudo labels generated by MaskCut on ImageNet train set. As mentioned in the original paper, its training stage takes 3 rounds where each round uses the detector of the previous round to infer on ImageNet train set as new pseudo labels.
- **UnSAM** (Wang et al., 2024): Its best setting is to train detectors on pseudo objects discovered by MaskCut on ImageNet train set for 3 rounds in the same way as CutLER. The final detector is used to infer on SA-1B train set. Another Mask2Former is trained on these pseudo labels.

- **CuVLER**: Its best setting is to first train a detector on pseudo labels generated by VoteCut on ImageNet train set, and then train a new detector on pseudo labels inferred from the trained detector on the COCO train set..
- **unMORE** (Ours): We just train a single detector on two groups of pseudo labels: one group from our discovered objects on COCO train set, another from object pseudo labels generated by VoteCut on ImageNet train set.

### 4.1. Multi-object Segmentation Results on COCO*

Table 1 shows **AP/AR** scores of all methods at different thresholds for object bounding boxes and masks.

**Results and Analysis of Methods in Group 1**: From rows 1-4 of Table 1, we can see that MaskCut and VoteCut which utilize DINO/v2 features can achieve preliminary performance. The middle block of Figure 5 shows qualitative results of MaskCut and VoteCut together with their used DINO/v2 features for grouping objects. Basically, these baselines mainly rely on grouping similar per-pixel features (obtained from pretrained DINO/v2) as objects, resulting in multiple similar objects being grouped as just one, as shown in Figure 5 where two cabinets are detected as one.

**Results and Analysis of Methods in Group 2**: From rows 5-7 of Table 1, we can see that our unMORE$_{disc}$ surpasses DINOSAUR and FOUND which are even inferior to feature similarity based methods in Group 1, meaning that reconstruction may not be a good object-centric grouping strategy and saliency maps may be misaligned with objectness.

Regarding our unMORE$_{disc}$, the right block of Figure 5 visualizes the learned center field and boundary distance

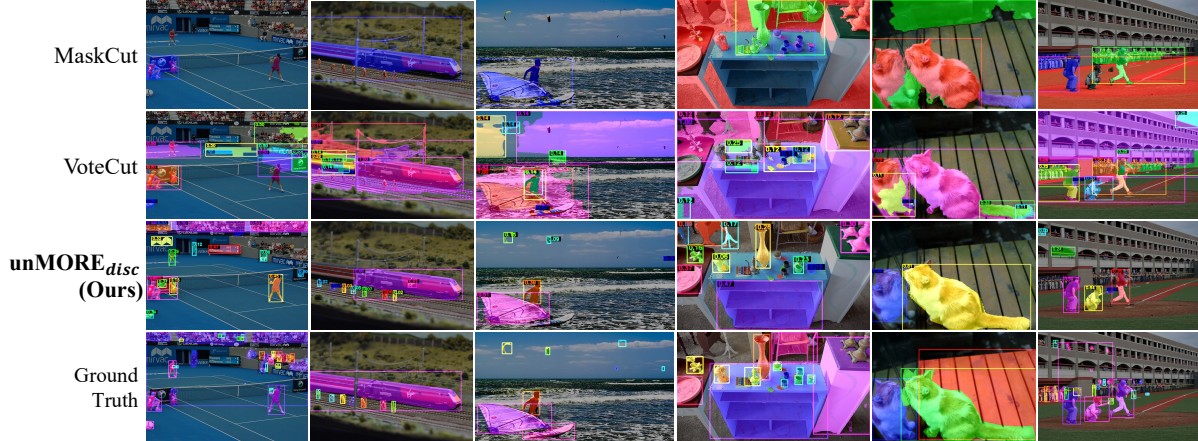

Figure 6: Qualitative results of Direct Object Discovery w/o CAD on COCO* val set as discussed in Sec 4.1 Groups 1&2.

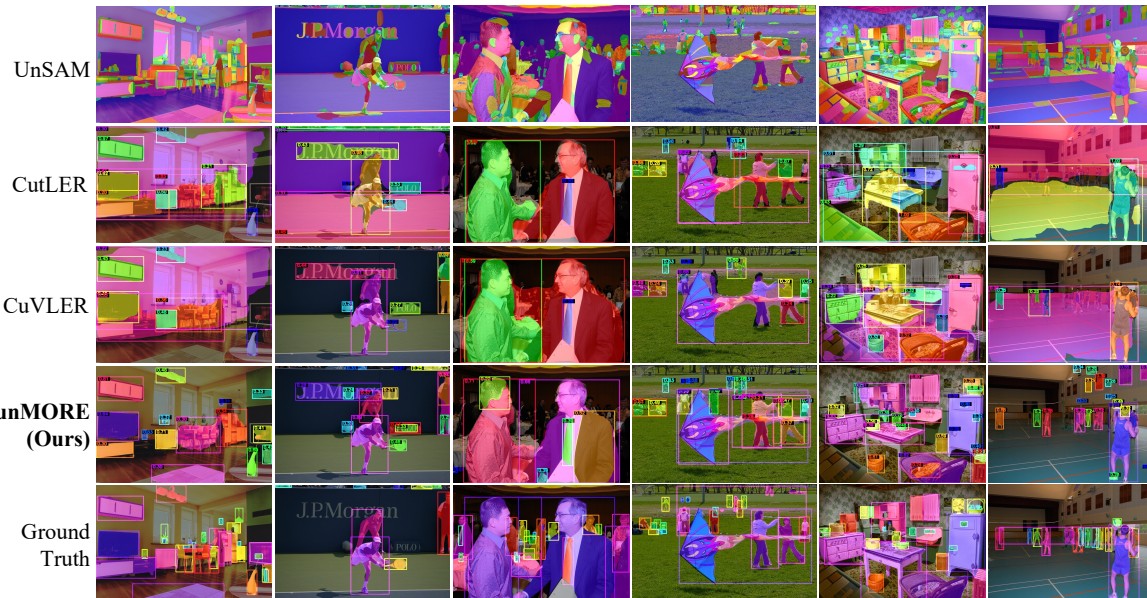

Figure 7: Qualitative results from trained detectors on COCO* val set as discussed in Sec 4.1 Group 3.

field, which allow us to easily discover individual objects, especially in crowded scenes. This is also verified by qualitative results presented in Figure 6. To further validate this insight, we separately calculate scores on images with more than 5/10/15 ground truth objects respectively in Table 5 of Appendix A.8. Our method maintains high scores on crowded images, whereas baselines collapse. Results on the original COCO val set (fewer annotations) are in Appendix A.10. More analysis is in Appendix A.17.

**Results and Analysis of Methods in Group 3**: From rows 8-11 of Table 1 and Figure 7, we can see that: 1) Our method clearly surpasses all methods by a large margin and achieves state-of-the-art performance. 2) Both CutLER and CuVLER can achieve reasonable results because additional detectors are likely to discover more objects. 3) The latest UnSAM appears to be incapable of identifying objects precisely, although it has a rather high AR score when its detector is

trained on the large-scale SA-1B dataset from SAM (Kirillov et al., 2023). Results on the original COCO validation set (fewer annotations) are provided in Appendix A.10.

## 4.2. Zero-shot Detection Results

For each method, we select its best-performing detector in Group 3 of Sec 4.1 and directly test it on another 6 datasets: COCO20K/ LVIS/ VOC/ KITTI/ Object365/ OpenImages. As shown in Table 2 and Figure 8, unMORE achieves the highest accuracy on all datasets across almost all metrics, showing our generalization in zero-shot detection.

We also note that, though our method achieves good performance for zero-shot detection on natural images, its capability is likely restricted by the learned objectness in training data. For data with significant domain gaps (*e.g.*, medical images), object priors from natural images may not apply.

Table 2: Quantitative results of zero-shot detection. Each method uses its best model in Group 3. Since KITTI/ VOC/ Object365/ OpenImages datasets do not have ground truth masks, only bounding box metrics are calculated.

| | COCO20K | | | | | | | | | | LVIS | | | | | | | | | |
|---|---|---|---|---|---|---|---|---|---|---|---|---|---|---|---|---|---|---|---|---|
| | $AP_{50}^{box}$ | $AP_{75}^{box}$ | $AP^{box}$ | $AR_{100}^{box}$ | $AR^{box}$ | $AP_{50}^{mask}$ | $AP_{75}^{mask}$ | $AP^{mask}$ | $AR_{100}^{mask}$ | $AR^{mask}$ | $AP_{50}^{box}$ | $AP_{75}^{box}$ | $AP^{box}$ | $AR_{100}^{box}$ | $AR^{box}$ | $AP_{50}^{mask}$ | $AP_{75}^{mask}$ | $AP^{mask}$ | $AR_{100}^{mask}$ | $AR^{mask}$ |
| UnSAM | 6.3 | 3.2 | 3.4 | 29.7 | **42.5** | 6.3 | 3.1 | 3.3 | 27.5 | **38.0** | 4.4 | 2.5 | 2.7 | 23.1 | **35.7** | 4.5 | 2.8 | 2.8 | 22.9 | **34.2** |
| CutLER | 22.4 | 11.9 | 12.5 | 33.1 | 33.1 | 19.6 | 9.2 | 10.0 | 27.2 | 27.2 | 8.5 | 3.9 | 4.5 | 21.8 | 21.8 | 6.7 | 3.2 | 3.5 | 18.7 | 18.7 |
| CuVLER | 24.1 | 12.3 | 13.1 | 32.6 | 32.6 | 21.1 | 9.7 | 10.7 | 27.2 | 27.2 | 8.9 | 4.1 | 4.7 | 20.8 | 20.8 | 7.2 | 3.4 | 3.8 | 17.9 | 17.9 |
| unMORE (Ours) | 25.9 | 13.0 | 13.9 | 35.4 | 35.4 | 23.6 | 11.1 | 12.0 | 30.5 | 30.5 | 10.4 | 5.0 | 5.6 | 24.1 | 24.1 | 8.9 | 4.5 | 4.9 | 21.4 | 21.4 |

| | KITTI | | | | | VOC | | | | | Object365 | | | | | OpenImages | | | | |
|---|---|---|---|---|---|---|---|---|---|---|---|---|---|---|---|---|---|---|---|---|
| | $AP_{50}^{box}$ | $AP_{75}^{box}$ | $AP^{box}$ | $AR_{100}^{box}$ | $AR^{box}$ | $AP_{50}^{box}$ | $AP_{75}^{box}$ | $AP^{box}$ | $AR_{100}^{box}$ | $AR^{box}$ | $AP_{50}^{box}$ | $AP_{75}^{box}$ | $AP^{box}$ | $AR_{100}^{box}$ | $AR^{box}$ | $AP_{50}^{box}$ | $AP_{75}^{box}$ | $AP^{box}$ | $AR_{100}^{box}$ | $AR^{box}$ |
| UnSAM | 1.9 | 0.6 | 0.8 | 17.0 | 21.7 | 5.1 | 2.3 | 2.6 | 38.8 | **51.9** | 9.1 | 4.9 | 5.3 | 30.5 | **47.9** | 6.6 | 3.7 | 4.0 | 34.6 | **48.7** |
| CutLER | 20.8 | 7.4 | 9.5 | 28.9 | 28.9 | 36.8 | 19.3 | 20.2 | 44.0 | 44.0 | 21.7 | 10.3 | 11.5 | 34.2 | 34.2 | 17.2 | 9.5 | 9.7 | 29.6 | 29.6 |
| CuVLER | 18.8 | 5.9 | 8.0 | 27.9 | 27.9 | 39.4 | 20.1 | 21.5 | 43.7 | 43.7 | 21.9 | 9.4 | 10.9 | 32.5 | 32.5 | 18.6 | **11.3** | **11.4** | 29.8 | 29.8 |
| unMORE (Ours) | 26.7 | 12.6 | 13.7 | 34.8 | 34.8 | 40.4 | 21.5 | 22.7 | 47.4 | 47.4 | 24.7 | 11.0 | 12.4 | 35.9 | 35.9 | 19.0 | 10.9 | 11.2 | 29.5 | 29.5 |

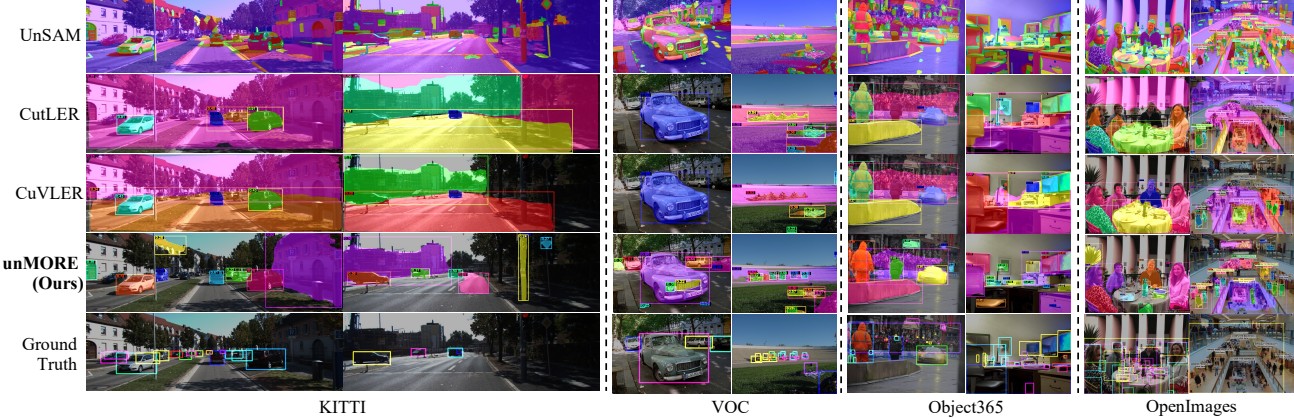

Figure 8: Qualitative results for zero-shot detection as discussed in Sec 4.2.

# 5. Ablations

We explore various combinations of these representations to train an objectness network, which then discovers objects as pseudo labels for the final detector. Details of ablation settings are in Appendix A.11.

Table 3: Ablation results of different choices for object-centric representations on COCO* val set.

| Object Existence | Object Center Field | Object Boundary Distance Field | $AP_{50}^{box}$ | $AP_{75}^{box}$ | $AP^{box}$ | $AR_{100}^{box}$ | $AR^{box}$ | $AP_{50}^{mask}$ | $AP_{75}^{mask}$ | $AP^{mask}$ | $AR_{100}^{mask}$ | $AR^{mask}$ |
|---|---|---|---|---|---|---|---|---|---|---|---|---|
| - | - | - | 23.4 | 10.7 | 11.8 | 33.8 | 33.8 | 19.6 | 8.0 | 9.4 | 35.7 | 35.7 |
| ✓ | - | - | 27.2 | 13.0 | 14.2 | 35.6 | 35.6 | 23.0 | 9.8 | 11.3 | 30.9 | 30.9 |
| - | ✓ | - | 29.2 | 14.9 | 15.8 | 37.3 | 37.3 | 25.6 | 11.8 | 13.0 | 32.5 | 32.5 |
| ✓ | ✓ | - | 29.0 | 14.4 | 15.4 | 36.3 | 36.3 | 25.0 | 11.1 | 12.5 | 31.0 | 31.0 |
| - | - | ✓ | 30.7 | 16.1 | 16.9 | 40.7 | 40.7 | 28.1 | 13.9 | 14.8 | 37.0 | 37.0 |
| ✓ | - | ✓ | 31.4 | 16.2 | 17.1 | 40.1 | 40.1 | 28.4 | 13.6 | 14.7 | 35.9 | 35.9 |
| - | ✓ | ✓ | 30.1 | 16.3 | 17.0 | 40.6 | 40.6 | 28.3 | 13.9 | 14.9 | 36.8 | 36.8 |
| ✓ | ✓ | ✓ | **32.6** | **17.2** | **18.0** | **40.9** | 40.9 | **29.6** | **14.4** | **15.5** | 36.5 | 36.5 |

With the above ablated versions, each method generates its pseudo labels on COCO train set. Then a detector is trained on these labels together with the same pseudo labels of ImageNet train set, exactly following the setting of our full method in Group 3 of Sec 4.1.

**Results & Analysis**: From Table 3, we can see that: 1) The boundary distance field yields the largest performance improvement, as it retains critical information of representing complex object boundaries, thus effectively helping discover more objects in the multi-object reasoning module. 2) Without learning object existence scores and object center fields, the AP score drops, potentially due to false positives

or under-segmentation in spite of a high AR score achieved. 3) The commonly used binary mask is far from sufficient to retain complex object-centric representations.

More ablations on our multi-object reasoning module, the choices of hyperparameters $\tau_{conf}^e$/$\tau_{conf}^c$/$\tau_{conf}^b$, and the data augmentation for objectness network are in Appendix A.12.

# 6. Conclusion

In this paper, we demonstrate that multiple objects can be accurately discovered from complex real-world images, without needing human annotations in training. This is achieved by our novel two-stage pipeline comprising an object-centric representation learning stage followed by a multi-object reasoning stage. We explicitly define three levels of object-centric representations to be learned from the large-scale ImageNet without human labels in the first stage. These representations serve as a key enabler for effectively discovering multi-objects on complex scene images in the second stage. Extensive experiments on multiple benchmarks demonstrate the state-of-the-art performance of our method in multi-object segmentation. It would be interesting to extend our framework to large-scale 2D image generation, where the large pretrained generative models may further improve the quality of object-centric representations.

## Acknowledgments

This work was supported in part by National Natural Science Foundation of China under Grant 62271431, in part by Research Grants Council of Hong Kong under Grants 25207822 & 15225522.

## Impact Statement

This paper presents work whose goal is to advance the field of Machine Learning. There are many potential societal consequences of our work, none which we feel must be specifically highlighted here.

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

# A. Appendix

The appendix includes:

## A.1. Details for Object-centric Representations

**Calculation of Signed Distance Field.** Given a binary mask $M \in \mathcal{R}^{H \times W \times 1}$, we calculate the distance from each pixel to its closest boundary point with `distanceTransform()` function in the `opencv` library (https://docs.opencv.org/4.x/d7/d1b/group__imgproc__misc.html). The function takes a binary mask as input and computes the shortest path length to the nearest zero pixel for all non-zero pixels. Thus, we first compute the distance field within the object, denoted as $S_{obj}$, using the object binary mask $M$. Then, we compute the distance field within the background, denoted as $S_{bg}$, using $(1 - M)$. The signed distance field for the whole image is $S = S_{obj} - S_{bg}$. Specifically, when using `distanceTransform()`, we set the distance type as L2 (Euclidean distance) and the mask size to be 3.

## A.2. Details for Objectness Network

**Objectness Network Architecture.** The *object existence* model employs ResNet50 (He et al., 2016) as the backbone. Following this backbone, the classification head consists of a single linear layer with output dimension of 1 and a sigmoid activation layer. The prediction for the *object center field* and the *object boundary distance* shares the same DPT-large (Ranftl et al., 2021) backbone with a 256-dimensional output size. Dense feature maps extracted from this backbone have the same resolution as input images and

the number of channels is 256. There are two prediction heads for the prediction of the *object center field* and the *object boundary distance*, respectively.

Table 4: Architecture of prediction heads for *object center field* and *object boundary distance*.

| center field prediction head | | | | boundary field prediction head | | | |
|---|---|---|---|---|---|---|---|
| type | channels | activation | stride | type | channels | activation | stride |
| layer 1 conv 1x1 | 512 | RELU | 1 | layer 1 conv 1x1 | 512 | RELU | 1 |
| layer 2 conv 3x3 | 512 | RELU | 1 | layer 2 conv 3x3 | 512 | RELU | 1 |
| layer 3 conv 1x1 | 1024 | RELU | 1 | layer 3 conv 1x1 | 1024 | RELU | 1 |
| layer 4 conv 1x1 | 2 | RELU | 1 | layer 4 conv 1x1 | 1 | RELU | 1 |

**Objectness Network Training Strategy.** The object existence model is trained using the Adam optimizer for 100K iterations with a batch size of 64. The learning rate is set to be a constant 0.0001. The object center and boundary models are jointly trained using the Adam optimizer for 50K iterations with a batch size of 16. The learning rate starts at 0.0001 and is divided by 10 at 10K and 20K iterations.

**Objectness Network Training Data.** We use the ImageNet train set with about 1.28 million images as the training set for the objectness network. For each ImageNet image, its object mask is the most confident mask generated by Vote-Cut proposed in CuVLER (Arica et al., 2024). For the training of the object existence model, negative samples that do not contain objects are created by cropping the largest rectangle region on the background. For positive samples that contain objects, we apply the random crop augmentation onto the original ImageNet image and discard the crop without a foreground object. For the training of the object center and boundary model, we first calculate the ground truth center field and boundary distance field based on the original full ImageNet image. Then, we apply the random crop augmentation onto the original image as well as the two representations. Specifically, the scale of the random crop is between 0.08 to 1, which implies the lower and upper bounds for the random area of the crop. The aspect ratio range of the random crop is between 0.75 and 1.33. Lastly, each image is resized to $128 \times 128$ before feeding into Objectness Network.

## A.3. Details for Multi-Object Reasoning Module

**Initial Object Proposal Generation.** Motivated by anchor box generation in Faster R-CNN (Ren et al., 2015). We use five scales $[32, 64, 128, 256, 512]$ and three aspect ratios $[0.5, 1, 2]$. At each scale, we randomly and uniformly sample proposal centers based on scale sizes. At each sampled center, we generate three boxes with different aspect ratios.

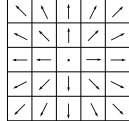

Figure 9: Predefined Kernel for Center Reasoning

**Predefined Kernel for Center Reasoning.** As illustrated

in Figure 9, each position within the kernel is defined as a 2-dimensional unit vector pointing towards the center of the kernel. Specifically, the value at the kernel center with position $[2, 2]$ is $(0, 0)$. The value at the $(i, j)^{th}$ position, denoted as $\mathbb{K}_{i,j}$, is defined and normalized as:

$$\mathbb{K}_{i,j} = \frac{[2, 2] - [i, j]}{\|[2, 2] - [i, j]\|}$$

To evaluate how *Center Field* matches with this anti-center pattern, we apply convolution onto *Center Field* with this kernel to calculate their average cosine similarity for each pixel in the *Center Field*. We set the threshold $\tau_c$ to be 0.25.

**More Details for Center Reasoning.** While deriving the *anti-center map* with the predefined kernel, we also find the boundary of the *Center Field*. Since on the *anti-center map*, values at the boundary of the *Center Field* will also be positive, we thus ignore the values on the *Center Field* boundary. Examples of center reasoning are provided in Figure 10.

**More Details for Boundary Reasoning.** Let $\boldsymbol{f}_p^b \in \mathcal{R}^{128 \times 128 \times 1}$ be the distance field for proposal $P$ and $\nabla \boldsymbol{f}_p^b \in \mathcal{R}^{128 \times 128 \times 2}$ is the gradient map for $\boldsymbol{f}_p^b$, where $\nabla \boldsymbol{f}_p^b[u, v] = (\frac{\partial \boldsymbol{f}_p^b}{\partial u}, \frac{\partial \boldsymbol{f}_p^b}{\partial v})$. And $\|\nabla \boldsymbol{f}_p^b\| \in \mathcal{R}^{128 \times 128 \times 1}$ is the norm for the gradient map. To make the bounding box update more stable, we use two strategies: (1) Use the averaged distance field gradient to replace the gradient at a single pixel position; (2) Apply adjustment on the calculated update step for a more aggressive expansion and conservative contraction.

(1) Since the distance field within the object and outside the object are normalized separately, the gradient average operation needs to be applied separately. Thus, we first apply sigmoid $\sigma$ function onto the boundary field to generate mask for foreground $\sigma(\boldsymbol{f}_p^b)$ and background $1 - \sigma(\boldsymbol{f}_p^b)$. Then gradients are averaged separately on the two masks and combined as the averaged gradient norm map for the distance field $AVG(\|\nabla \boldsymbol{f}_p^b\|) \in \mathcal{R}^{128 \times 128 \times 1}$. We replace $\|\nabla \boldsymbol{f}_p^b\|$ with $AVG(\|\nabla \boldsymbol{f}_p^b\|)$ when calculating box updates.

$$AVG(\|\nabla \boldsymbol{f}_p^b\|) = \frac{\sum \sigma(\boldsymbol{f}_p^b) \cdot \|\nabla \boldsymbol{f}_p^b\|}{\sum \sigma(\boldsymbol{f}_p^b)} \cdot \sigma(\boldsymbol{f}_p^b) \quad (7)$$

$$+ \frac{\sum (1 - \sigma(\boldsymbol{f}_p^b)) \cdot \|\nabla \boldsymbol{f}_p^b\|}{\sum (1 - \sigma(\boldsymbol{f}_p^b))} \cdot (1 - \sigma(\boldsymbol{f}_p^b)) \quad (8)$$

(2) Empirically, box contraction needs to be more conservative since objects could be overlooked if the proposal is over-tightened. For example, for a person wearing a tie, if the proposal around the person gets shrunk too much, the object of interest may transfer to the tie instead. Also, for efficiency, it is suitable to make more aggressive expansion

since objects can still be well seen from a proposal larger than its tightest bounding box. Thus, we further adjust the calculated updates with an adjustment ratio $\tau_{adjust} = 0.5$. Instead of directly using Eq. 6, we use the following formulas to calculate boundary update:

$$P^{u_1} \longleftarrow P^{u_1} - \frac{max(\boldsymbol{f}_{p_t}^b)}{\|\frac{\partial \boldsymbol{f}_{p_t}^b}{\partial u}, \frac{\partial \boldsymbol{f}_{p_t}^b}{\partial v}\|} - \tau_{adjust} * \frac{\|max(\boldsymbol{f}_{p_t}^b)\|}{\|\frac{\partial \boldsymbol{f}_{p_t}^b}{\partial u}, \frac{\partial \boldsymbol{f}_{p_t}^b}{\partial v}\|},$$
$$where \ (u, v) = argmax \boldsymbol{f}_{p_t}^b \quad (9)$$

$$P^{v_1} \longleftarrow P^{v_1} - \frac{max(\boldsymbol{f}_{p_l}^b)}{\|\frac{\partial \boldsymbol{f}_{p_l}^b}{\partial u}, \frac{\partial \boldsymbol{f}_{p_l}^b}{\partial v}\|} - \tau_{adjust} * \frac{\|max(\boldsymbol{f}_{p_l}^b)\|}{\|\frac{\partial \boldsymbol{f}_{p_l}^b}{\partial u}, \frac{\partial \boldsymbol{f}_{p_l}^b}{\partial v}\|},$$
$$where \ (u, v) = argmax \boldsymbol{f}_{p_l}^b \quad (10)$$

$$P^{u_2} \longleftarrow P^{u_2} + \frac{max(\boldsymbol{f}_{p_b}^b)}{\|\frac{\partial \boldsymbol{f}_{p_b}^b}{\partial u}, \frac{\partial \boldsymbol{f}_{p_b}^b}{\partial v}\|} + \tau_{adjust} * \frac{\|max(\boldsymbol{f}_{p_b}^b)\|}{\|\frac{\partial \boldsymbol{f}_{p_b}^b}{\partial u}, \frac{\partial \boldsymbol{f}_{p_b}^b}{\partial v}\|},$$
$$where \ (u, v) = argmax \boldsymbol{f}_{p_b}^b \quad (11)$$

$$P^{v_2} \longleftarrow P^{v_2} + \frac{max(\boldsymbol{f}_{p_r}^b)}{\|\frac{\partial \boldsymbol{f}_{p_r}^b}{\partial u}, \frac{\partial \boldsymbol{f}_{p_r}^b}{\partial v}\|} + \tau_{adjust} * \frac{\|max(\boldsymbol{f}_{p_r}^b)\|}{\|\frac{\partial \boldsymbol{f}_{p_r}^b}{\partial u}, \frac{\partial \boldsymbol{f}_{p_r}^b}{\partial v}\|},$$
$$where \ (u, v) = argmax \boldsymbol{f}_{p_r}^b \quad (12)$$

**Parameters for Proposal Updating.** Each proposal undergoes at most 50 iterations of updates. For efficiency, we stop a proposal from being updated once it meets the following criteria. Specifically, the calculated maximum expansion for the proposal should be smaller than 0 (it means the border moves outside of the object boundary), and the maximum shrinkage should be smaller than a small margin, which we set to be 16 pixels. While it is acceptable for the proposal to be slightly larger than the tightest bounding box, it should not be smaller. Examples of boundary reasoning can be found in Figure 10.

### A.4. Details for Object Mask and Confidence Score Calculation

For a converged proposal $P$, we can compute its object mask $M_p$ as the union of mask from center field and mask from boundary field:

$$M_p^{center} = \begin{cases} 1, & \text{if } \|f_p^c\| \geq 0.5 \\ 0, & \text{otherwise} \end{cases} \quad (13)$$

$$M_p^{boundary} = \begin{cases} 1, & \text{if } \sigma(\boldsymbol{f}_p^b) \geq 0.5 \\ 0, & \text{otherwise} \end{cases} \quad (14)$$

$$M_p = \cup(M_p^{center}, M_p^{boundary}) \quad (15)$$

To calculate the confidence score $conf_p$ for proposal $P$, we consider its object existence score, center field, and boundary field. Specifically, we also consider mask area when calculating the confidence by comparing the object area in $P$ with other objects' areas within the same image. Suppose there are $K$ discovered objects within the image, the final score is calculated as:

$$conf_p = f_p^e * max(\|\boldsymbol{f}_p^c\|) * max(\boldsymbol{f}_p^b) * \left( \frac{\sum \boldsymbol{M}_p}{max_{k \in K} \sum \boldsymbol{M}_k} \right)^{0.25} \tag{16}$$

### A.5. Details for Pseudo Label Processing

Given a set of discovered objects from scene images, we perform selection and assign each of them a weight to use them as pseudo labels for training the detector. Following the definition in the Section A.4, an object proposal $P$ will be selected if it satisfies three conditions below:

$$f_p^e \geq \tau_{conf}^e; \; max(\|\boldsymbol{f}_p^c\|) \geq \tau_{conf}^c; \; max(\boldsymbol{f}_p^b) \geq \tau_{conf}^b \tag{17}$$

The three thresholds correspond to object existence score ($\tau_{conf}^e$), maximum norm in *center field* ($\tau_{conf}^c$) and maximum value in *boundary distance field* ($\tau_{conf}^b$). In our paper, we set:

$$\tau_{conf}^e = 0.5; \quad \tau_{conf}^c = 0.8; \quad \tau_{conf}^b = 0.75 \tag{18}$$

For each selected proposal, its weight for the detector training is determined by its relative area in the scene image: $\left( \frac{\sum \boldsymbol{M}_p}{max_{k \in K} \sum \boldsymbol{M}_k} \right)^{0.25}$.

### A.6. Details for Detector Training

The architecture for the Class Agnostic Detector is Cascade Mask RCNN. All experiments are performed with the Detectron2 (Wu et al., 2019) platform. Detectors are optimized for 25K iterations using SGD optimizer with a learning rate of 0.005 and a batch size of 16. We use a weight decay of 0.00005 and 0.9 momentum. Following CutLER (Wang et al., 2023a), we also use copy-paste augmentation with a uniformly sampled downsample ratio between 0.3 and 1.0.

### A.7. Details for Datasets

**COCO** (Lin et al., 2014): The MS COCO (Microsoft Common Objects in Context) dataset is a large-scale object detection and segmentation dataset. The COCO in the paper refers to the 2017 version that contains 118K training images and 5K validation images.

**COCO 20K** (Lin et al., 2014): COCO 20K is a subset of the COCO trainval2014 with 19817 images. Since it contains images from both training and validation set from the 2014

version of COCO, this dataset is generally used to evaluate unsupervised approaches.

**LVIS** (Gupta et al., 2019): LVIS (Large Vocabulary Instance Segmentation) is a dataset for long tail instance segmentation. It contains 164,000 images with more than 1,200 categories and more than 2 million high-quality instance-level segmentation masks.

**KITTI** (Geiger et al., 2012): KITTI (Karlsruhe Institute of Technology and Toyota Technological Institute) is one of the most popular datasets for use in mobile robotics and autonomous driving. Our method is evaluated with 7521 images from its trainval split.

**PASCAL VOC** (Everingham et al., 2010): The PASCAL Visual Object Classes (VOC) 2012 dataset is a widely used benchmark for object detection, containing 1464 training images and 1449 validation images.

**Object365 V2** (Shao et al., 2019): Objects365 is a large-scale object detection dataset. It has 365 object categories and over 600K training images. We evaluate our method in terms of object detection on its validation split with 80K images.

**OpenImages V6** (Kuznetsova et al., 2020): OpenImages V6 is a large-scale dataset, consists of 9 million training images, 41,620 validation samples, and 125,456 test samples. We evaluate our method in terms of object detection on its validation split.

### A.8. More Results on COCO* Validation Set

We present a detailed evaluation on COCO* validation dataset based on object count in Table 5. We can see that, when the number of objects in each image is rather small (*e.g.*, [0 - 4]), the results of top-performing baselines Vote-Cut/CuVLER are comparable to our method, all yielding high scores. However, as the number of objects per image increases (*e.g.*, $\geq 5$ objects), our unMORE$_{disc}$/ unMORE consistently outperforms all baselines by growing margins, demonstrating the superiority of our method in dealing with challenging crowded images.

Notably, UnSAM achieves high $AR^{box}$/$AR^{mask}$ scores (used in the original UnSAM paper to measure the average recall rate without limiting the number of predictions), but its $AR_{100}^{box}$/$AR_{100}^{mask}$ scores (only considers the top 100 predictions per image and commonly adopted for object segmentation) are clearly lower. This is because UnSAM focuses on excessively partitioning images by clustering granular segments, which sacrifices the accuracy of object discovery, but tends to oversegment objects. This is also qualitatively validated in the Figure 7 and Figure 8 in the main paper.

Table 5: Quantitative results on COCO* validation set based on object count. "# of GT obj." refers to the average number of ground truth objects per image, while "# of pred obj." refers to that of predicted objects.

| # of GT obj. | | Direct Object Discovery | | | | | | | | | | | Training Detectors | | | | | | | | | | |
|---|---|---|---|---|---|---|---|---|---|---|---|---|---|---|---|---|---|---|---|---|---|---|---|
| | | $AP^{box}_{50}$ | $AP^{box}_{75}$ | $AP^{box}$ | $AR^{box}_{100}$ | $AR^{box}$ | $AP^{mask}_{50}$ | $AP^{mask}_{75}$ | $AP^{mask}$ | $AR^{mask}_{100}$ | $AR^{mask}$ | # pred obj | | $AP^{box}_{50}$ | $AP^{box}_{75}$ | $AP^{box}$ | $AR^{box}_{100}$ | $AR^{box}$ | $AP^{mask}_{50}$ | $AP^{mask}_{75}$ | $AP^{mask}$ | $AR^{mask}_{100}$ | $AR^{mask}$ | # pred obj |
| [0,4] | MaskCut (K=3) | 25.1 | 12.3 | 13.3 | 28.5 | 28.5 | 22.5 | 8.9 | 10.6 | 24.2 | 24.2 | 1.8 | UnSAM | 15.5 | 10.5 | 10.2 | 66.4 | 73.3 | 15.9 | 10.4 | 10.1 | 60.1 | 65.5 | 244.1 |
| | MaskCut (K=10) | 24.5 | 11.7 | 12.9 | 29.3 | 29.3 | 21.9 | 8.8 | 10.3 | 24.8 | 24.8 | 1.9 | CutLER | 55.2 | 35.4 | 34.4 | 61.2 | 61.2 | 51.2 | 29.0 | 28.6 | 52.4 | 52.4 | 100.0 |
| | VoteCut | 38.9 | 21.1 | 22.0 | **39.1** | **39.1** | 37.0 | 17.4 | 19.0 | 34.3 | 34.3 | 8.5 | CuVLER | **56.9** | **36.1** | **35.1** | 60.5 | 60.5 | **53.6** | **30.1** | **29.9** | 52.6 | 52.6 | 99.9 |
| | **unMORE_{disc} (Ours)** | **42.1** | **21.2** | **23.2** | 38.7 | 38.7 | **42.1** | **22.0** | **22.8** | **37.8** | **37.8** | 5.9 | **unMORE (Ours)** | 55.3 | 33.7 | 33.1 | 59.6 | 59.6 | 52.9 | 29.5 | 29.5 | **52.8** | **52.8** | 100.0 |
| [5,9] | MaskCut (K=3) | 10.7 | 4.8 | 5.3 | 10.4 | 10.4 | 9.4 | 3.4 | 4.3 | 9.0 | 9.0 | 1.9 | UnSAM | 13.3 | 8.6 | 8.7 | **49.9** | **62.0** | 13.5 | 8.5 | 8.5 | 46.2 | **56.3** | 317.9 |
| | MaskCut (K=10) | 11.4 | 5.0 | 5.5 | 11.4 | 11.4 | 9.6 | 3.6 | 4.4 | 9.9 | 9.9 | 2.1 | CutLER | 37.7 | 21.4 | 21.7 | 49.3 | 49.3 | 33.3 | 16..7 | 17.5 | 42.5 | 42.5 | 100.0 |
| | VoteCut | 17.2 | 7.6 | 8.6 | 17.5 | 17.5 | 15.4 | 6.7 | 7.4 | 15.0 | 15.0 | 9.0 | CuVLER | 39.0 | 21.1 | 21.9 | 48.4 | 48.4 | 34.1 | 16.6 | 17.8 | 41.3 | 41.3 | 99.6 |
| | **unMORE_{disc} (Ours)** | **25.2** | **12.7** | **13.7** | **25.8** | **25.8** | **24.0** | **12.3** | **12.8** | **24.2** | **24.2** | 7.9 | **unMORE (Ours)** | **40.8** | **21.8** | **22.8** | 49.9 | 49.9 | **37.0** | **18.6** | **19.6** | **44.4** | 44.4 | 100.0 |
| [10,14] | MaskCut (K=3) | 5.1 | 2.3 | 2.7 | 4.9 | 4.9 | 4.4 | 1.6 | 1.8 | 4.3 | 4.3 | 1.9 | UnSAM | 11.2 | 6.7 | 6.9 | 38.5 | **52.7** | 11.3 | 6.7 | 6.8 | 36.6 | **48.6** | 378.1 |
| | MaskCut (K=10) | 5.4 | 2.4 | 2.7 | 5.5 | 5.5 | 4.7 | 1.4 | 1.9 | 4.8 | 4.8 | 2.3 | CutLER | 26.3 | 13.2 | 14.3 | 40.3 | 40.3 | 22.8 | 10.2 | 11.5 | 34.9 | 34.9 | 100.0 |
| | VoteCut | 8.8 | 3.0 | 3.9 | 9.5 | 9.5 | 7.2 | 3.6 | 3.3 | 8.1 | 8.1 | 9.2 | CuVLER | 28.5 | 13.7 | 15.2 | 39.7 | 39.7 | 24.9 | 11.1 | 12.4 | 34.0 | 34.0 | 99.7 |
| | **unMORE_{disc} (Ours)** | **18.0** | **8.2** | **9.4** | **19.3** | **19.3** | **16.7** | **7.7** | **8.6** | **18.5** | **18.5** | 9.4 | **unMORE (Ours)** | **33.4** | **16.7** | **17.9** | **43.2** | 43.2 | **30.5** | **14.3** | **15.7** | **38.6** | 38.6 | 100.0 |
| [15, +) | MaskCut (K=3) | 1.8 | 0.5 | 0.7 | 1.9 | 1.9 | 1.6 | 0.4 | 0.6 | 1.6 | 1.6 | 2.0 | UnSAM | 8.9 | 5.2 | 5.5 | 24.8 | **40.9** | 8.6 | 4.9 | 5.2 | 24.2 | **38.2** | 475.7 |
| | MaskCut (K=10) | 1.7 | 0.5 | 0.8 | 2.1 | 2.1 | 1.5 | 0.4 | 0.7 | 1.9 | 1.9 | 2.3 | CutLER | 18.3 | 9.0 | 10.0 | 29.0 | 29.0 | 15.5 | 6.7 | 7.6 | 25.1 | 25.1 | 100.0 |
| | VoteCut | 4.2 | 1.4 | 1.8 | 4.6 | 4.6 | 3.2 | 1.2 | 1.4 | 4.0 | 4.0 | 9.3 | CuVLER | 21.4 | 9.7 | 10.9 | 28.4 | 28.4 | 17.0 | 7.1 | 8.3 | 24.4 | 24.4 | 99.6 |
| | **unMORE_{disc} (Ours)** | **13.6** | **6.5** | **7.1** | **14.0** | **14.0** | **12.3** | **5.6** | **6.3** | **13.4** | **13.4** | 12.2 | **unMORE (Ours)** | **29.0** | **14.2** | **15.3** | **33.4** | 33.4 | **24.8** | **11.0** | **12.5** | **30.0** | 30.0 | 100.0 |

## A.9. Details for CAD Training Settings

In Sec 4.1 Group 3, since four methods train CAD with different settings, we adopt a diverse range of training settings, which are detailed as follows. The best setting for each method is marked with **bold**. Full results for all settings on COCO* validation set are in Table 6.

1) For UnSAM, it has two detectors trained under two settings below. Both models are from the original paper and are included for reference.

- Setting #1: It trains a detector on pseudo objects discovered by MaskCut on ImageNet train set, and then the detector is used to infer scene images jointly with Mask-Cut.
- **Setting #2**: The detector trained in its Setting #1 is used to infer pseudo objects on SA-1B train set. Another Mask2Former is trained on these pseudo labels for inference on scene images.

2) For CutLER, it has three detectors trained under three settings below. The Settings #1/#2 are fairly comparable with our Settings #1/#2, whereas its Setting #3 is from the original paper.

- Setting #1: It is trained on pseudo objects discovered by its own MaskCut on COCO train set.
- Setting #2: It is trained on two groups of pseudo labels: one group from its discovered objects on COCO train set, another from object pseudo labels generated by MaskCut on ImageNet train set.
- **Setting #3**: It is trained on object pseudo labels generated by MaskCut on ImageNet train set.

3) For CuVLER, it has four detectors trained under four settings below. The Settings #1/#2 are fairly comparable with our Settings #1/#2, whereas its Settings #3/#4 are from the original paper.

- Setting #1: It is trained only on pseudo objects discovered by its own VoteCut on COCO train set.

- Setting #2: It is trained on two groups of pseudo labels: one group from its discovered objects on COCO train set, another from object pseudo labels generated by VoteCut on ImageNet train set.
- Setting #3: It is trained only on object pseudo labels generated by VoteCut on ImageNet train set.
- **Setting #4**: It first uses the detector of Setting #3 to infer object pseudo labels on COCO train set, and then trains a new detector on these pseudo labels.

4) For our method, named unMORE, we train two separate detectors under two settings:

- Setting #1: It is trained only on pseudo objects discovered by our method on COCO train set.
- **Setting #2**: It is trained on two groups of pseudo labels: one group from our discovered objects on COCO train set, another from object pseudo labels generated by VoteCut on ImageNet train set.

## A.10. Results on the Original COCO Validation Set

This section presents the experiment results evaluated on original COCO validation set. Table 7 shows the quantitative results on COCO validation set. Table 8 shows quantitative results of detectors with different settings on the original COCO validation set.

## A.11. Details for Ablation Settings

As mentioned in Sec 5, We explore various combinations of these representations to train objectness network, which then discovers objects as pseudo labels for the final detector. Details of ablation settings are as follows:

**1) Only using a binary mask as the object-centric representation**: In the task of object segmentation, a binary mask is probably the most commonly-used object representation. In particular, we remove all of our three object-centric representations, but just train the same objectness network to predict a binary mask. Then, when discovering multi-

Table 6: Quantitative results of detectors with different settings on COCO* validation set.

| | Training Settings | $AP_{50}^{box}$ | $AP_{75}^{box}$ | $AP^{box}$ | $AR_{100}^{box}$ | $AP_{50}^{mask}$ | $AP_{75}^{mask}$ | $AP^{mask}$ | $AR_{100}^{mask}$ |
|---|---|---|---|---|---|---|---|---|---|
| UnSAM | Setting #1 | 3.5 | 2.1 | 2.3 | 30.5 | 3.2 | 2.0 | 2.1 | 27.2 |
| | Setting #2 | 10.2 | 6.3 | 6.4 | 36.1 | 10.2 | 6.2 | 6.3 | 34.1 |
| CutLER | Setting #1 | 21.2 | 10.8 | 11.6 | 33.4 | 18.2 | 8.1 | 9.1 | 27.7 |
| | Setting #2 | 23.6 | 11.8 | 12.6 | 33.7 | 19.8 | 8.3 | 9.5 | 28.4 |
| | Setting #3 | 26.0 | 14.2 | 14.7 | 37.9 | 22.7 | 11.2 | 11.8 | 32.7 |
| CuVLER | Setting #1 | 26.1 | 13.2 | 14.1 | 36.0 | 22.6 | 10.3 | 11.3 | 30.6 |
| | Setting #2 | 27.0 | 13.0 | 14.2 | 35.0 | 23.2 | 10.1 | 11.4 | 29.8 |
| | Setting #3 | 27.2 | 14.0 | 14.9 | 37.2 | 23.2 | 10.7 | 11.8 | 30.2 |
| | Setting #4 | 28.0 | 14.8 | 15.5 | 37.8 | 24.4 | 11.7 | 12.6 | 32.1 |
| **unMORE (Ours)** | Setting #1 | 31.2 | 15.6 | 16.8 | 40.0 | 28.8 | 12.7 | 14.9 | 36.1 |
| | Setting #2 | **32.6** | **17.2** | **18.0** | **40.9** | **29.6** | **14.4** | **15.5** | **36.5** |

Table 7: Quantitative results on the original COCO validation dataset.

| | | | Trainable Module | $AP_{50}^{box}$ | $AP_{75}^{box}$ | $AP^{box}$ | $AR_{100}^{box}$ | $AR^{box}$ | $AP_{50}^{mask}$ | $AP_{75}^{mask}$ | $AP^{mask}$ | $AR_{100}^{mask}$ | $AR^{mask}$ | avg. # obj. |
|---|---|---|---|---|---|---|---|---|---|---|---|---|---|---|
| Direct Object Discovery | w/o Learnable Modules | FreeMask | - | 4.1 | 0.7 | 1.4 | 4.3 | 4.3 | 3.5 | 0.4 | 1.1 | 3.4 | 3.4 | 3.7 |
| | | MaskCut (K=3) | - | 6.4 | 2.5 | 3.1 | 7.7 | 7.7 | 5.4 | 1.8 | 2.3 | 6.5 | 6.5 | 1.8 |
| | | MaskCut (K=10) | - | 6.0 | 2.7 | 3.1 | 8.2 | 8.2 | 5.5 | 1.7 | 2.2 | 6.9 | 6.9 | 2.1 |
| | | VoteCut | - | 11.0 | 5.0 | 5.6 | 12.4 | 12.4 | 9.4 | 4.0 | 4.6 | 10.5 | 10.5 | 8.9 |
| | w/o Learnable Modules | DINOSAUR | Recon. SlotAtt | 2.1 | 0.2 | 0.6 | 5.5 | 5.5 | 0.8 | 0.1 | 0.2 | 2.5 | 2.5 | 7.0 |
| | | FOUND | Seg. Head | 4.7 | 2.1 | 2.3 | 4.5 | 4.5 | 3.7 | 1.5 | 1.8 | 3.7 | 3.7 | 1.0 |
| | | **unMORE$_{disc}$ (Ours)** | Obj. Net | 15.7 | 6.9 | 7.9 | 16.5 | 16.5 | 14.7 | 6.9 | 7.5 | 15.9 | 15.9 | 8.2 |
| Training Detectors | - | UnSAM | Detector x 4 | 5.9 | 3.2 | 3.4 | 30.0 | **42.4** | 5.9 | 3.1 | 3.3 | 27.4 | **37.9** | 332.2 |
| | | CutLER | Detector x 3 | 22.9 | 11.7 | 12.4 | 31.8 | 31.8 | 18.7 | 7.3 | 8.8 | 23.9 | 23.9 | 100.0 |
| | | CuVLER | Detector x 2 | 23.4 | 12.1 | 12.8 | 32.2 | 32.2 | 20.4 | 9.6 | 10.4 | 26.8 | 26.8 | 99.7 |
| | | **unMORE (Ours)** | **Obj. Network + Detector x 1** | **25.4** | **12.7** | **13.6** | **35.2** | 35.2 | **22.9** | **10.7** | **11.7** | **30.3** | 30.3 | 100.0 |

objects on scene images, we manually set a suitable step size to extensively search object candidates by querying the pretrained network.

**2) Only using a binary mask and an object existence score**: This is to evaluate whether the object existence score can be useful for better object segmentation. In the absence of object boundary field, the binary mask representation can update bounding boxes.

**3) Only using a binary mask and an object center field**: This is to evaluate whether the object center field can be useful for better object segmentation. In the absence of object boundary field, the binary mask representation can update bounding boxes.

**4) Using a binary mask, an object existence score and center field**: This is to evaluate whether both object existence score and center field can be useful for better object segmentation. In the absence of object boundary field, the binary mask representation can update bounding boxes.

**5) Only using an object boundary field**: This is to verify the importance of object boundary field.

**6) Only using an object boundary field and existence score**: This is to evaluate whether adding the existence score can help object segmentation on top of the object boundary field.

**7) Only using an object boundary field and center field**: This is to evaluate whether adding the center field can help object segmentation on top of the object boundary field.

**8) Our full three-level object-centric representations**: This is our full framework for reference.

### A.12. More Ablations

**Selection of Fixed Step Size for Binary Baseline.** Since the information provided by binary mask representation is very limited, the final discovered objects can be very sensitive to the step size. In order to choose a good step size in favor of the binary mask baseline, we randomly select 100 images from COCO* validation set and evaluate the results for a step size of 5, 15, 20, 30. According to the results shown in Table 9, we select 20 as the fixed step size.

**Ablation on Parameters for Pseudo Label Processing.** We perform ablation studies on the parameters used in A.5. Specifically, we choose a wide range, i.e., $(0 \sim 0.95)$ for score thresholds of object existence $\tau_{conf}^{e}$, object center $\tau_{conf}^{c}$ and object boundary $\tau_{conf}^{b}$ on 7 datasets. As shown in Tables 10&11, more tolerant thresholds lead to higher AR scores because more objects can be discovered, but a decrease in AP because of low-quality detections. On the other hand, if thresholds are too strict, both AR and AP scores drop because only a limited number of objects are discovered. Nevertheless, our method is not particularly sensitive to the selection of thresholds as it demonstrates good performance across different thresholds.

**Ablation on Random Cropping Augmentation for the Objectness Network**. During training our objectness network on ImageNet, we originally apply random cropping

Table 8: Quantitative results of detectors with different settings on the original COCO validation set.

| | Training Settings | $AP_{50}^{box}$ | $AP_{75}^{box}$ | $AP^{box}$ | $AR_{100}^{box}$ | $AP_{50}^{mask}$ | $AP_{75}^{mask}$ | $AP^{mask}$ | $AR_{100}^{mask}$ |
|---|---|---|---|---|---|---|---|---|---|
| UnSAM | Setting #1 | 2.1 | 1.1 | 1.2 | 27.0 | 1.8 | 0.9 | 1.0 | 23.5 |
| | Setting #2 | 5.9 | 3.2 | 3.4 | 30.0 | 5.9 | 3.1 | 3.3 | 27.4 |
| CutLER | Setting #1 | 19.3 | 9.9 | 10.6 | 29.4 | 16.3 | 7.3 | 8.2 | 23.2 |
| | Setting #2 | 20.8 | 10.4 | 11.1 | 29.7 | 17.2 | 7.0 | 8.1 | 23.3 |
| | Setting #3 | 21.9 | 11.8 | 12.3 | 32.7 | 18.9 | 9.2 | 9.7 | 27.0 |
| CuVLER | Setting #1 | 22.9 | 11.7 | 12.4 | 31.8 | 18.7 | 7.3 | 8.8 | 23.9 |
| | Setting #2 | 23.2 | 11.3 | 12.3 | 31.2 | 19.7 | 8.5 | 9.5 | 24.9 |
| | Setting #3 | 22.9 | 11.8 | 12.6 | 32.9 | 19.3 | 8.9 | 9.8 | 25.1 |
| | Setting #4 | 23.4 | 12.1 | 12.8 | 32.2 | 20.4 | 9.6 | 10.4 | 26.8 |
| **unMORE (Ours)** | Setting #1 | 24.1 | 11.2 | 12.5 | 34.2 | 22.2 | 9.9 | 11.1 | 29.9 |
| | Setting #2 | **25.4** | **12.7** | **13.6** | **35.2** | **22.9** | **10.7** | **11.7** | **30.3** |

Table 9: Results of different step sizes for binary baseline on COCO* validation set.

| step size | $AP_{50}^{box}$ | $AP_{75}^{box}$ | $AP^{box}$ | $AR_{100}^{box}$ | $AP_{50}^{mask}$ | $AP_{75}^{mask}$ | $AP^{mask}$ | $AR_{100}^{mask}$ |
|---|---|---|---|---|---|---|---|---|
| 5 | 10.4 | 4.1 | 4.9 | 10.8 | 9.4 | 3.7 | 4.3 | 9.6 |
| 15 | 12.4 | 5.8 | 6.4 | 12.1 | 10.7 | 4.8 | 5.4 | **10.7** |
| **20** | **13.1** | **6.3** | **6.9** | **12.1** | **11.6** | **5.3** | **5.9** | **10.7** |
| 30 | 11.8 | 5.4 | 6.1 | 11.5 | 10.1 | 4.6 | 5.1 | 10.1 |

augmentation. Here, we conduct an additional ablation study by omitting the random cropping operation during training the objectness network while keeping all other settings the same. Table 12 shows the quantitative results on the COCO* validation set. We can see that random cropping is indeed helpful for the objectness network to learn robust center and boundary fields. Primarily, this is because during the multi-object reasoning stage, many proposals just have partial or fragmented objects, but the random cropping augmentation inherently enables the objectness network to infer rather accurate center and boundary field for those partial objects, thus driving the proposals to be updated correctly.

**Ablation on Rough Masks for Training Objectness Network.** We conducted the following ablation study on four types of pseudo-masks:

- SelfMask (Shin et al., 2022b): For each image, we employ the strong unsupervised saliency detection model SelfMask to predict a salient region as the pseudo label.
- MaskCut: For each image, we use the first object discovered by MaskCut as the pseudo label.
- VoteCut: It's used in our main paper.
- VoteCut+SAM: For each image, a rough mask is generated by VoteCut, and its bounding box is used as a prompt for SAM to predict the final pseudo mask. While this yields the best pseudo labels, SAM is a fully supervised model, so this ablation is for reference only.

As shown in Table 13, our method is amenable to all types of rough masks, though their quality affects $unMORE_{disc}$ performance. While SAM scores highest, its improvement over VoteCut is not substantial, as it still relies on bounding box prompts from VoteCut. Importantly, our method does not depend on specific pretrained features, enabling the use of enhanced pretrained models in the future.

### A.13. Time Consumption and Throughput

Time consumption is summarized in Table 14. $unMORE_{disc}$ takes 10 hours to train the objectness network and is slower for Direct Object Discovery. However, our subsequent detector unMORE requires only 30 hours to train, benefiting from the high-quality pseudo labels from $unMORE_{disc}$, while baseline detectors take over 60 hours. Ultimately, the inference speed of our unMORE matches that of CutLER and CuVLER.

Regarding the throughput, for each image on average, the number of initial proposals is 1122.7, whereas the number of predicted objects from $unMORE_{disc}$ is 8.9. Most initial proposals have low existence scores and are discarded at the first iteration. The Non-Maximum Suppression (NMS) will also remove redundant proposals.

### A.14. Failure Cases

We present failure cases in Figure 11 and discuss limitations as follows.

1. Direct Object Discovery of $unMORE_{disc}$ takes time. It could be possible to leverage reinforcement learning techniques to learn an efficient policy net to discover objects.

2. Our method struggles to separate overlapping objects with similar textures, as shown in the attached Figure 11. Additional language priors may help alleviate this issue.

### A.15. More Visualizations

Table 10: Ablation results for thresholds of object existence $\tau_{conf}^e$, object center $\tau_{conf}^c$ and object boundary $\tau_{conf}^b$ on COCO* validation set.

| $\tau_{conf}^e$ | $\tau_{conf}^c$ | $\tau_{conf}^b$ | $AP_{50}^{box}$ | $AP_{75}^{box}$ | $AP^{box}$ | $AR_{100}^{box}$ | $AP_{50}^{mask}$ | $AP_{75}^{mask}$ | $AP^{mask}$ | $AR_{100}^{mask}$ |
|---|---|---|---|---|---|---|---|---|---|---|
| **0.0** | 0.8 | 0.75 | 31.2 | 16.7 | 17.4 | **41.0** | 28.7 | **14.6** | 15.3 | **37.2** |
| **0.25** | 0.8 | 0.75 | 31.5 | 16.7 | 17.5 | 40.8 | 28.6 | 14.3 | 15.2 | 36.7 |
| 0.5 | 0.8 | 0.75 | **32.6** | **17.2** | **18.0** | 40.9 | **29.6** | 14.4 | **15.5** | 36.5 |
| **0.75** | 0.8 | 0.75 | 30.8 | 16.2 | 16.9 | 38.9 | 27.7 | 13.3 | 14.3 | 34.7 |
| **0.95** | 0.8 | 0.75 | 28.1 | 13.4 | 14.7 | 34.4 | 24.3 | 10.7 | 12.1 | 30.1 |
| 0.5 | **0.0** | 0.75 | 32.5 | 16.4 | 17.5 | 40.0 | 29.2 | 13.6 | 14.9 | 35.8 |
| 0.5 | **0.25** | 0.75 | 31.8 | 16.4 | 17.3 | 39.9 | 28.5 | 13.5 | 14.7 | 35.7 |
| 0.5 | **0.5** | 0.75 | 31.0 | 16.2 | 17.0 | 40.2 | 27.7 | 13.3 | 14.4 | 36.0 |
| 0.5 | 0.8 | 0.75 | **32.6** | **17.2** | **18.0** | 40.9 | 29.6 | 14.4 | 15.5 | **36.5** |
| 0.5 | **0.95** | 0.75 | 29.8 | 15.8 | 16.5 | 38.1 | 26.8 | 13.2 | 14.1 | 34.2 |
| 0.5 | 0.8 | **0.0** | 31.8 | 16.0 | 17.0 | 38.7 | 28.4 | 13.2 | 14.5 | 34.6 |
| 0.5 | 0.8 | **0.25** | 31.2 | 16.1 | 17.0 | 38.9 | 27.8 | 13.2 | 14.3 | 34.7 |
| 0.5 | 0.8 | **0.5** | 31.7 | 16.9 | 17.5 | 40.6 | 28.4 | 13.7 | 14.7 | 36.0 |
| 0.5 | 0.8 | 0.75 | **32.6** | **17.2** | **18.0** | **40.9** | **29.6** | **14.4** | **15.5** | **36.5** |
| 0.5 | 0.8 | **0.95** | 31.6 | 17.5 | 17.9 | 39.8 | 28.0 | 13.3 | 14.5 | 35.0 |

Table 11: Ablation results for thresholds of object existence $\tau_{conf}^e$, object center $\tau_{conf}^c$ and object boundary $\tau_{conf}^b$ on COCO20K, LVIS, KITTI, VOC, Object365 and OpenImages.

| | | | COCO | | | | COCO20K | | | | LVIS | | | | KITTI | | VOC | | Object365 | | OpenImages | |
|---|---|---|---|---|---|---|---|---|---|---|---|---|---|---|---|---|---|---|---|---|---|---|
| $\tau_{conf}^e$ | $\tau_{conf}^c$ | $\tau_{conf}^b$ | $AP_{50}^{box}$ | $AR_{100}^{box}$ | $AP_{50}^{mask}$ | $AR_{100}^{mask}$ | $AP_{50}^{box}$ | $AR_{100}^{box}$ | $AP_{50}^{mask}$ | $AR_{100}^{mask}$ | $AP_{50}^{box}$ | $AR_{100}^{box}$ | $AP_{50}^{mask}$ | $AR_{100}^{mask}$ | $AP_{50}^{box}$ | $AR_{100}^{box}$ | $AP_{50}^{box}$ | $AR_{100}^{box}$ | $AP_{50}^{box}$ | $AR_{100}^{box}$ | $AP_{50}^{box}$ | $AR_{100}^{box}$ |
| **0.0** | 0.8 | 0.75 | 23.8 | 35.1 | 21.9 | **30.8** | 24.3 | 35.2 | 22.6 | **31.1** | 10.2 | **24.9** | 9.0 | 22.6 | 25.3 | 32.5 | 38.5 | 46.9 | 23.6 | **36.3** | 18.3 | **29.5** |
| **0.25** | 0.8 | 0.75 | 24.1 | 34.8 | 22.0 | 30.3 | 24.6 | 35.0 | 22.6 | 30.6 | 10.2 | 24.4 | 8.7 | 21.9 | 25.0 | 34.0 | 39.1 | 46.6 | 23.8 | 36.0 | 18.7 | 29.4 |
| 0.5 | 0.8 | 0.75 | **25.4** | 35.2 | 22.9 | 30.3 | 25.9 | 35.4 | 23.6 | 30.5 | **10.4** | 24.1 | 8.9 | 21.4 | **26.7** | **34.8** | 40.4 | 47.4 | 24.7 | 35.9 | **19.0** | 29.5 |
| **0.75** | 0.8 | 0.75 | 24.5 | 33.7 | 21.9 | 28.8 | 25.1 | 34.1 | 22.7 | 29.2 | 9.9 | 22.5 | 8.3 | 20.0 | 25.5 | 33.6 | 40.4 | 46.7 | 23.8 | 36.0 | 18.7 | 29.4 |
| **0.95** | 0.8 | 0.75 | 23.2 | 30.2 | 19.9 | 25.0 | 23.8 | 30.5 | 20.6 | 25.3 | 8.7 | 18.8 | 6.9 | 16.3 | 21.6 | 29.6 | 39.4 | 43.7 | 21.6 | 30.0 | 18.8 | 26.5 |
| 0.5 | **0.0** | 0.75 | **25.7** | 34.5 | 22.8 | 29.8 | **26.2** | 34.8 | 23.4 | 30.1 | **10.4** | 23.3 | 8.5 | 20.9 | **28.7** | **35.5** | 41.3 | 47.0 | 24.5 | 35.1 | 19.7 | 29.0 |
| 0.5 | **0.25** | 0.75 | 25.0 | 34.4 | 22.2 | 29.5 | 25.6 | 34.8 | 23.0 | 29.8 | 10.1 | 23.2 | 8.3 | 20.6 | 27.7 | 33.6 | 41.0 | 46.8 | 23.8 | 35.1 | 19.3 | 29.0 |
| 0.5 | **0.5** | 0.75 | 24.5 | 34.7 | 21.8 | 29.9 | 25.1 | 34.8 | 22.5 | 30.1 | 9.8 | 23.6 | 8.0 | 21.1 | 24.1 | 32.7 | 40.3 | 46.7 | 23.3 | 35.3 | **19.9** | **29.7** |
| 0.5 | 0.8 | 0.75 | 25.4 | **35.2** | 22.9 | **30.3** | 25.9 | **35.4** | 23.6 | **30.5** | **10.4** | 24.1 | **8.9** | 21.4 | 26.7 | 34.8 | 40.4 | **47.4** | **24.7** | **35.9** | 19.0 | 29.5 |
| 0.5 | **0.95** | 0.75 | 23.7 | 32.9 | 21.1 | 28.3 | 24.3 | 33.2 | 21.8 | 28.5 | 9.6 | 21.6 | 8.2 | 19.3 | 25.7 | 33.3 | 38.6 | 45.6 | 22.5 | 33.2 | 18.3 | 28.4 |
| 0.5 | 0.8 | **0.0** | 24.7 | 33.4 | 21.9 | 28.7 | 25.3 | 33.6 | 22.6 | 29.0 | 10.1 | 22.3 | 8.2 | 19.8 | 27.4 | 33.4 | 40.0 | 45.9 | 23.6 | 33.8 | 19.3 | 28.3 |
| 0.5 | 0.8 | **0.25** | 24.6 | 33.6 | 21.8 | 28.9 | 25.3 | 34.0 | 22.5 | 29.3 | 9.8 | 22.4 | 8.0 | 19.8 | 26.7 | 33.5 | 40.7 | 46.1 | 23.2 | 34.1 | 19.7 | 28.6 |
| 0.5 | 0.8 | **0.5** | 25.3 | **35.2** | 22.4 | 30.0 | **25.9** | 35.3 | 23.1 | 30.4 | 10.0 | 23.6 | 8.4 | 20.9 | 25.4 | 34.3 | **41.3** | **47.8** | 23.7 | 35.8 | **19.9** | **29.9** |
| 0.5 | 0.8 | 0.75 | **25.4** | 35.2 | **22.9** | 30.3 | 25.9 | **35.4** | **23.6** | **30.5** | 10.4 | 24.1 | 8.9 | **21.4** | 26.7 | 34.8 | 40.4 | 47.4 | **24.7** | **35.9** | 19.0 | 29.5 |
| 0.5 | 0.8 | **0.95** | 20.4 | 32.2 | 19.7 | 28.6 | 24.4 | 34.4 | 22.7 | 29.8 | **10.5** | 23.3 | **9.0** | 21.0 | **29.7** | **35.1** | 37.6 | 46.4 | 23.8 | 34.8 | 17.8 | 29.2 |

Table 12: Ablation results on COCO* validation set for random cropping augmentation of the objectness network.

| | $AP_{50}^{box}$ | $AP_{75}^{box}$ | $AP^{box}$ | $AR_{100}^{box}$ | $AP_{50}^{mask}$ | $AP_{75}^{mask}$ | $AP^{mask}$ | $AR_{100}^{mask}$ |
|---|---|---|---|---|---|---|---|---|
| unMORE$_{disc}$ (with random cropping) | 19.1 | 9.0 | 10.1 | 19.6 | 17.8 | 8.7 | 9.5 | 18.9 |
| unMORE$_{disc}$ (w/o random cropping) | 15.7 | 7.5 | 8.2 | 18.1 | 15.6 | 6.6 | 7.9 | 17.4 |

Table 13: Ablation study for rough masks.

| | Rough Masks | SSL features / Supervision | $AP_{50}^{box}$ | $AP_{75}^{box}$ | $AP^{box}$ | $AR_{100}^{box}$ | $AR^{box}$ | $AP_{50}^{mask}$ | $AP_{75}^{mask}$ | $AP^{mask}$ | $AR_{100}^{mask}$ | $AR^{mask}$ |
|---|---|---|---|---|---|---|---|---|---|---|---|---|
| unMORE$_{disc}$ | SelfMask | DINO_b16, MoCov2, SwAV | 13.2 | 6.1 | 4.8 | 16.4 | 16.4 | 12.0 | 5.0 | 5.6 | 15.3 | 15.3 |
| | MaskCut | DINO_b8 | 16.3 | 7.3 | 6.4 | 17.7 | 17.7 | 14.3 | 5.7 | 6.1 | 18.7 | 18.7 |
| | VoteCut | DINO_b8, DINO_s8, DINO_b16, DINO_s16, DINOv2_s14, DINOv2_b14 | 19.1 | 9.0 | 10.1 | 19.6 | 19.6 | 17.8 | 8.7 | 9.5 | 18.9 | 18.9 |
| | VoteCut + SAM | supervised on SA-1B dataset | **21.9** | **9.1** | **10.7** | **19.7** | **19.7** | **18.4** | **9.2** | **9.9** | **19.1** | **19.1** |

Table 14: Training and inference time of different methods. For a fair comparison, all methods are evaluated on the same hardware configurations.

| | Training Time (hours in total) | | | | Inference Efficiency (seconds per image) | | | |
|---|---|---|---|---|---|---|---|---|
| Direct Object Discovery | MaskCut (N=3) | MaskCut (N=10) | VoteCut | unMORE$_{disc}$ | MaskCut (N=3) | MaskCut (N=10) | VoteCut | unMORE$_{disc}$ |
| | - | - | - | 10.1 | 11.3 | 33.7 | 5.1 | 45.3 |
| Training Detectors | UnSAM | CutLER | CuVLER | unMORE | UnSAM | CutLER | CuVLER | unMORE |
| | 90.0 | 75.0 | 60.0 | 30.0 | 3.0 | 0.1 | 0.1 | 0.1 |

**Multi-Object Reasoning Overview**

Center Reasoning → Split Proposals → Boundary Reasoning → Iteration 0 / Iteration 2 / Iteration 4

**Center Reasoning Details**

Proposal Splitting ✂

**Boundary Reasoning Details**

Proposal #1

Proposal #2

Proposal #3

Proposal #4

Proposal #5

Proposal #6

Iteration 0 / Iteration 2 / Iteration 4

Figure 10: Multi-object reasoning with object center and boundary representations on a multi-object image.

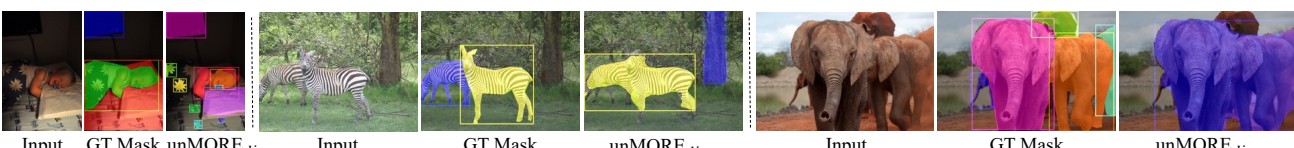

Input   GT Mask   unMORE$_{disc}$      Input   GT Mask   unMORE$_{disc}$      Input   GT Mask   unMORE$_{disc}$

Figure 11: Failure cases of unMORE$_{disc}$.

### A.16. Details of COCO* Validation Set

In COCO*, we exhaustively label objects in the COCO val2017 dataset, which comprises 5,000 images and originally contains 36,781 instances across 90 categories. We have added 197 new object categories and labeled previously unannotated objects within the original COCO categories. In total, COCO* includes 5,000 images, 287 categories, and 47,117 labeled objects. Details for the annotated categories are provided in Table 15. We use SAM (Kirillov et al., 2023) to expedite the labeling process. We label each object of interest with a tightest bounding box around it. This bounding box, along with the full image, is then fed into the SAM model to generate a dense binary mask.

Table 15: Details of COCO* validation set. This table includes the unique class IDs, class names and the number of newly labeled objects that belong to each class. Specifically, the newly introduced classes are assigned with IDs from 100 to 297. Apart from the 197 new categories, we also label objects belonging to the original COCO classes (the id between 1-90) that are not labeled in COCO validation 2017. In summary, we have labeled 10,336 objects in addition to the original 36,781 objects on COCO validation 2017, resulting in 47,117 objects on 5,000 images.

| id | class name | count | id | class name | count | id | class name | count | id | class name | count |
|---|---|---|---|---|---|---|---|---|---|---|---|
| 3 | car | 9 | 128 | tissue | 184 | 183 | cabbage | 24 | 247 | corn | 9 |
| 11 | fire hydrant | 1 | 129 | rice | 27 | 184 | cucumber | 39 | 248 | plum | 5 |
| 15 | bench | 6 | 130 | painting | 445 | 185 | calendar | 13 | 249 | MP3 player | 6 |
| 17 | cat | 2 | 131 | board | 40 | 186 | pinapple | 19 | 250 | garlic | 3 |
| 20 | sheep | 3 | 132 | ballon | 49 | 187 | key | 11 | 251 | scallion | 2 |
| 33 | suitcase | 1 | 133 | camera | 71 | 188 | pumpkin | 6 | 252 | noodle | 9 |
| 44 | bottle | 175 | 134 | handler | 73 | 189 | ball | 15 | 253 | soup | 14 |
| 47 | cup | 44 | 135 | soap | 19 | 190 | calculator | 6 | 254 | onion | 6 |
| 49 | knife | 5 | 136 | brush | 37 | 191 | flashlight | 8 | 255 | sausage | 20 |
| 50 | spoon | 8 | 137 | shower | 21 | 192 | usb | 13 | 256 | vegatable | 19 |
| 51 | bowl | 17 | 138 | beetroot | 6 | 193 | potato | 15 | 257 | fishbowl | 4 |
| 53 | apple | 19 | 139 | meat | 102 | 194 | ipad | 5 | 258 | wallet | 3 |
| 56 | broccoli | 1 | 140 | bridge | 11 | 195 | pad | 40 | 259 | buoy | 15 |
| 57 | carrot | 11 | 141 | grape | 55 | 196 | banner | 174 | 260 | roadblock | 56 |
| 59 | pizza | 4 | 142 | cheese | 10 | 197 | funnel | 3 | 261 | chocolate | 12 |
| 61 | cake | 12 | 143 | clothes | 102 | 198 | blender | 30 | 262 | shell | 7 |
| 62 | chair | 34 | 144 | box | 186 | 199 | name tag | 125 | 263 | wool | 5 |
| 63 | couch | 2 | 145 | curtain | 228 | 200 | jar | 74 | 264 | avocado | 1 |
| 67 | dining table | 2 | 146 | beans | 15 | 201 | flag | 156 | 265 | charger | 9 |
| 70 | toilet | 10 | 147 | dustbin | 131 | 202 | peach | 4 | 266 | card | 4 |
| 75 | remote | 1 | 148 | broom | 6 | 203 | radio | 5 | 267 | coin | 4 |
| 76 | keyboard | 63 | 149 | stand | 86 | 204 | helmet | 466 | 268 | wire | 9 |
| 77 | cell phone | 4 | 150 | statue | 69 | 205 | cart | 32 | 269 | piano | 6 |
| 79 | oven | 11 | 151 | fries | 16 | 206 | toothpaste | 14 | 270 | chinaware | 13 |
| 81 | sink | 35 | 152 | plastic bag | 104 | 207 | coconut | 6 | 271 | balance | 2 |
| 82 | refrigerator | 1 | 153 | blanket | 71 | 208 | salmon | 21 | 272 | pancake | 3 |
| 84 | book | 18 | 154 | bathtub | 38 | 209 | tongs | 1 | 273 | pepper | 8 |
| 86 | vase | 16 | 155 | stationary | 59 | 210 | CD player | 34 | 274 | eggplant | 2 |
| 101 | cabinet | 291 | 156 | sauce | 47 | 211 | heater | 18 | 275 | napkin | 18 |
| 102 | carpet | 65 | 157 | poster | 194 | 212 | air conditioner | 12 | 276 | table stand | 3 |
| 103 | lamp | 495 | 158 | sail | 5 | 213 | butterfly | 22 | 277 | kiwifruit | 1 |
| 104 | basket | 87 | 159 | rhino | 3 | 214 | tent | 15 | 278 | fig | 1 |
| 105 | pillow | 312 | 160 | paper | 142 | 215 | salad | 18 | 279 | soother | 2 |
| 106 | mirror | 67 | 161 | hook | 28 | 216 | spagatti | 6 | 280 | pomelo | 2 |
| 107 | pot | 227 | 162 | hand dryer | 1 | 217 | gravestone | 9 | 281 | guita | 2 |
| 108 | hat | 179 | 163 | tomato | 53 | 218 | arcade game machine | 1 | 282 | screen | 15 |
| 109 | scarf | 13 | 164 | lemon | 18 | 219 | chips | 12 | 283 | callbox | 2 |
| 110 | flower | 253 | 165 | snail | 1 | 220 | fish | 16 | 284 | map | 4 |
| 111 | applicance | 82 | 166 | candle | 70 | 221 | pig | 1 | 285 | coffee machine | 1 |
| 112 | can | 71 | 167 | teapot | 46 | 222 | dish | 71 | 286 | dishwasher | 1 |
| 113 | skate shoe | 189 | 168 | moon | 4 | 223 | CD | 30 | 287 | soap stand | 1 |
| 114 | glove | 143 | 169 | strawberry | 26 | 224 | doll | 29 | 288 | shelf | 12 |
| 115 | stove | 45 | 170 | paperbag | 20 | 225 | watermelon | 6 | 289 | prize | 0 |
| 116 | watch | 38 | 171 | lid | 30 | 226 | cherry | 4 | 290 | tower | 5 |
| 117 | ornament | 187 | 172 | earphone | 32 | 227 | cream | 12 | 291 | picture | 13 |
| 118 | oar | 4 | 173 | egg | 28 | 228 | toy | 43 | 292 | vent | 5 |
| 119 | speaker | 90 | 174 | butter | 10 | 229 | pomegranate | 1 | 293 | baggage tag | 32 |
| 120 | printer | 22 | 175 | tap | 220 | 230 | rolling pin | 2 | 294 | biscuit | 7 |
| 121 | monitor | 4 | 176 | fan | 38 | 231 | envolop | 3 | 295 | telescope | 1 |
| 122 | basin | 75 | 177 | switch | 128 | 241 | sticker | 51 | 296 | pear | 5 |
| 123 | road sign | 555 | 178 | telephone | 34 | 242 | dough | 7 | 297 | ferris wheel | 2 |
| 124 | towel | 213 | 179 | socket | 114 | 243 | pan | 12 | 298 | lizard | 1 |
| 125 | ashtray | 7 | 180 | bag | 86 | 244 | peanut | 1 | | | |
| 126 | plate | 190 | 181 | quilt | 46 | 245 | billboard | 154 | | | |
| 127 | bread | 87 | 182 | tank | 11 | 246 | ladder | 6 | | | |

### A.17. More Results and Analysis of Object-centric Representations

In this section, we provide more insights into the comparison between our proposed center-boundary representations and self-supervised features. In particular, we experiment with four pre-trained models from DINO and two pre-trained models from DINOv2, with different patch sizes and/or model parameter scales.

Motivated by NCut (Shi & Malik, 2000) algorithm, given a set of image features, we construct a weighted graph. The weight on each edge is computed as the similarity between features, formulating an affinity matrix $W$. Then, we solve an eigenvalue system $(D - W)x = \lambda Dx$ for a set of eigenvectors $x$ and eigenvalues $\lambda$, where D is the diagonal matrix. In Figures 12 & 13 & 14 & 15 & 16, we visualize the eigenvectors corresponding to the 2nd, 3rd, and 4th smallest eigenvalues. Specifically, we resize all eigenvectors to be the same size as the source image.

In practice, methods like TokenCut (Wang et al., 2023b) and CuVLER (Arica et al., 2024) directly use the eigenvector corresponding to the 2nd smallest eigenvalue and perform clustering onto it.

From Figures 12 & 13 & 14 & 15 & 16, we observe that segmenting objects via grouping pre-trained self-supervised features: 1) focuses on large objects that dominate the image, while ignoring objects with smaller sizes, 2) tends to capture semantic similarity / background-foreground contrast, instead of objectness. For example, in Figure 12, only the "bed" object with a large size can be discovered by clustering eigenvectors. In Figure 13, the two "keyboards", two "monitors", and two "speakers" are hard to be distinguished into separate clusters. Such behaviors are fundamentally due to the training of self-supervised features only involving image-level contrast, which can hardly lead to fine-grained object understanding.

In contrast, as shown in the last row of Figures 12 & 13 & 14 & 15 & 16, our proposed center and boundary representation captures more fine-grained properties that directly reflect objectness, which naturally leads to better object discovery results. It should be noted that the merged center field and merged boundary distance field are derived by combining all proposals with their predicted center field and boundary distance field, instead of predicted in one pass.

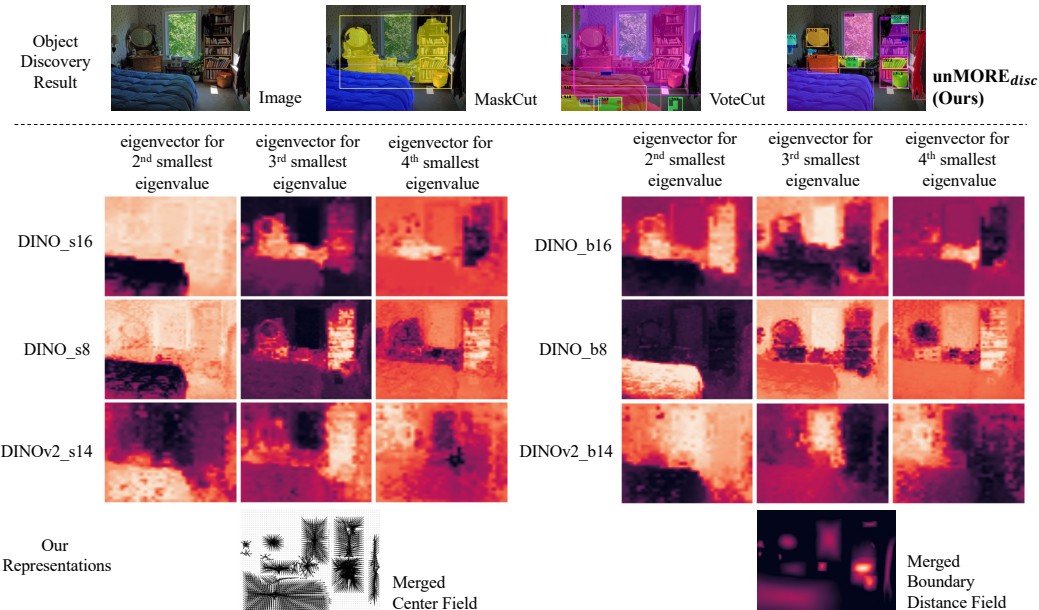

Figure 12: Comparison between DINO/DINOv2 features with proposed boundary-center representations. The eigenvectors are reshaped to be the size of the image. The last row shows the illustrations for the proposed center and boundary distance representations (predicted).

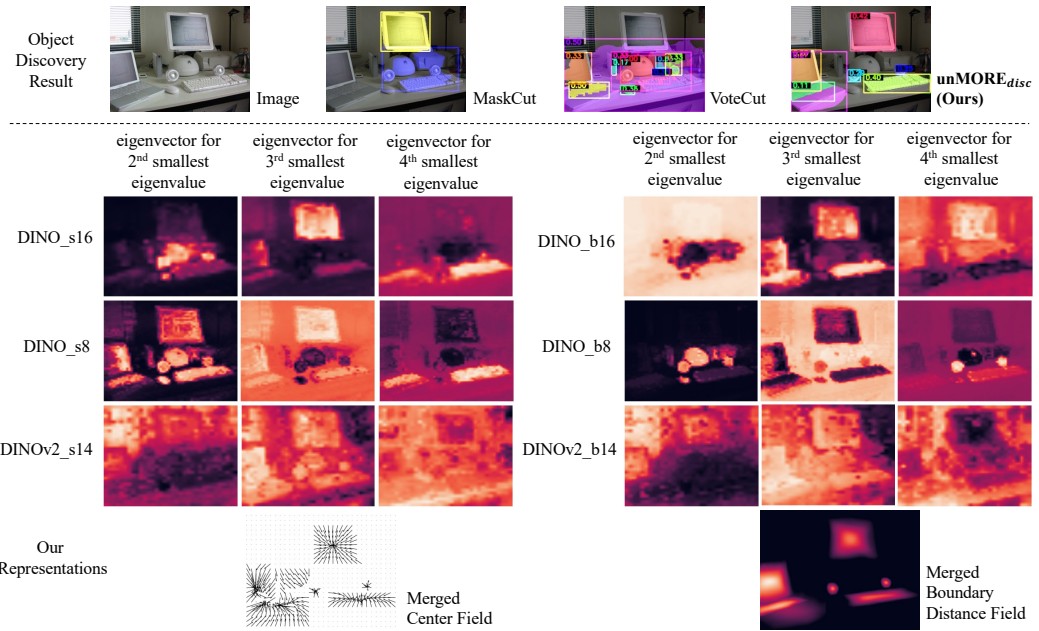

Figure 13: Comparison between DINO/DINOv2 features with proposed boundary-center representations. The eigenvectors are reshaped to be the size of the image. The last row shows the illustrations for the proposed center and boundary distance representations (predicted).

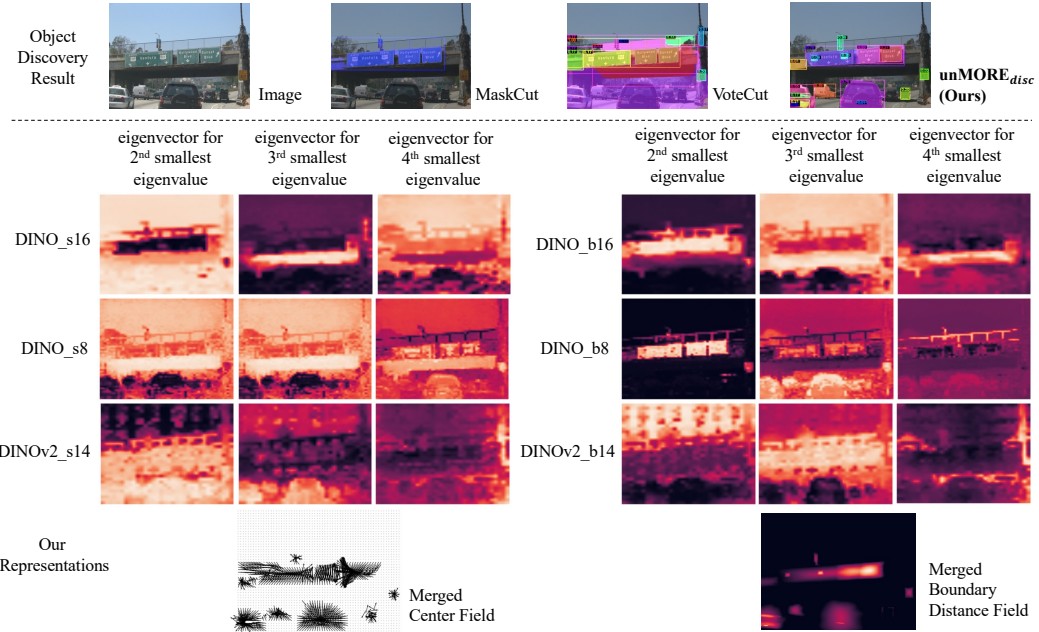

Figure 14: Comparison between DINO/DINOv2 features with proposed boundary-center representations. The eigenvectors are reshaped to be the size of the image. The last row shows the illustrations for the proposed center and boundary distance representations (predicted).

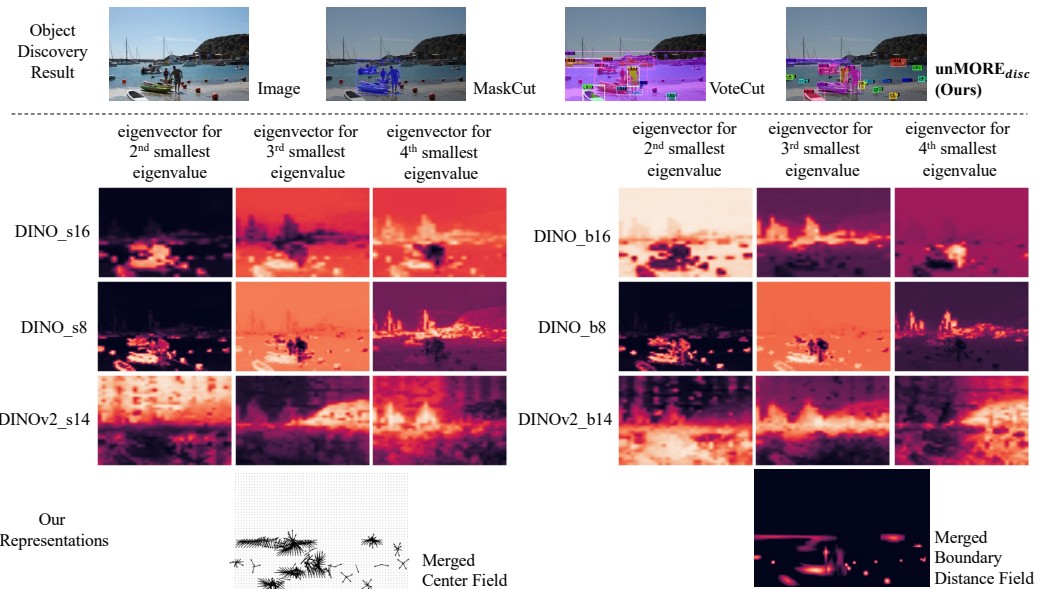

Figure 15: Comparison between DINO/DINOv2 features with proposed boundary-center representations. The eigenvectors are reshaped to be the size of the image. The last row shows the illustrations for the proposed center and boundary distance representations (predicted).

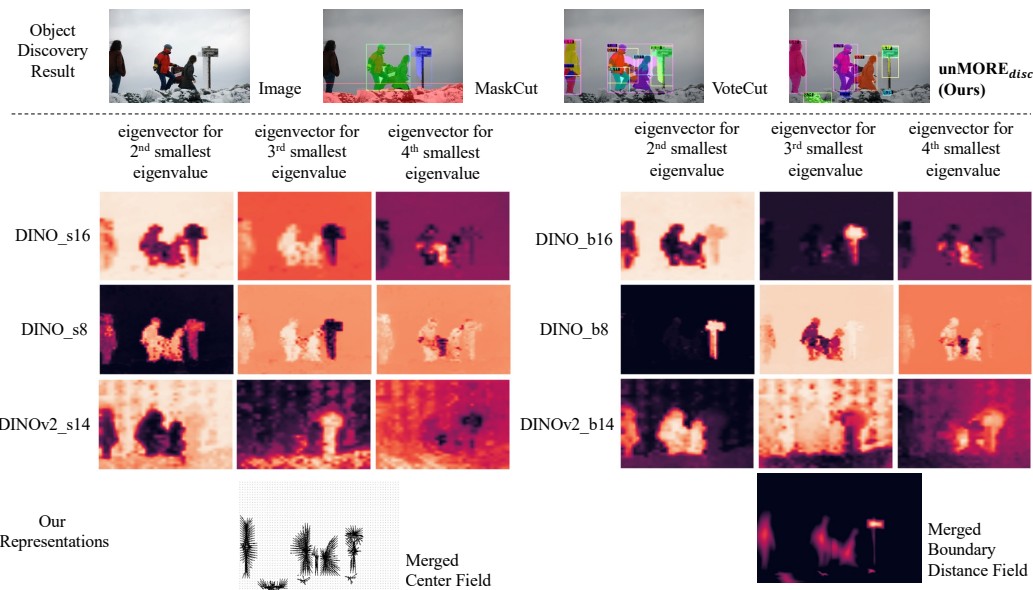

Figure 16: Comparison between DINO/DINOv2 features with proposed boundary-center representations. The eigenvectors are reshaped to be the size of the image. The last row shows the illustrations for the proposed center and boundary distance representations (predicted).

### A.18. Efficiency of Direct Object Discovery

For our method of direct object discovery on the COCO* validation set as described in Group 2 of Sec 4.1, in implementation, the maximum number of iterations to optimize a proposal is set to be 50. Nevertheless, in practice, as shown in Figure 17 which illustrates the relationship between the average number of pixels to increase or decrease and the number of optimization steps, we observe that all proposals tend to converge after just 10 iterations.

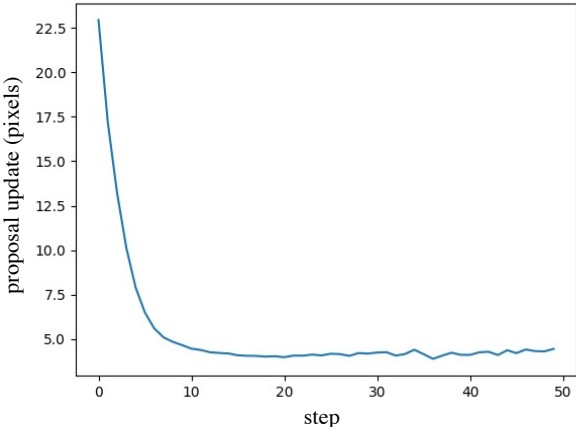

Figure 17: The relationship between the average number of pixels to increase/decrease and the number of optimization steps.

