# OpenReview forum: "unMORE: Unsupervised Multi-Object Segmentation via Center-Boundary Reasoning"
_ICML.cc/2025/Conference — ICML 2025 poster_

### Official Review · Reviewer_CHsG · 2025-03-12

**Overall Recommendation:** 3

**Summary:**

The paper proposes a novel framework for unsupervised object segmentations. It proposes a two stage solution by incorporating an objectiveness network that is trained on an object centric dataset (ImageNet) to predict the existence, location, and boundary of each object, and a reasoning module to generate final predictions based on several heuristics. Experiments show that the proposed methods outperforms existing baselines on COCO and SA-1B dataset.

**Claims And Evidence:**

The paper's main claim was well-supported with empirical performance gains on a wide range  of benchmarks and metrics, suggesting the proposed method is effective

**Essential References Not Discussed:**

N/A

**Experimental Designs Or Analyses:**

The author provide a set of comprehensive ablation

**Methods And Evaluation Criteria:**

The author conducted experiments on many common benchmarks such as COCO, SA-1B, LIVIS following the standard practice of the community. The choices of benchmarks are sound.

**Other Comments Or Suggestions:**

I'm curious about how authors chose the selected metrics for reporting. It seems the results are mostly focused on AP, and only include AR_100. This setup largely follows Cutler. However, it appears the authors of UnSAM focused more on AR. I'm wondering what is the performance on AR_mask (not AR_mask_100) on COCO. In particular, Table 1 of UnSAM paper showed that their results (measured by AR_mask)  is only marginally behind the supervised SAM baseline. Since this work offers considerable gain over UnSAM, it may be able to further reduce the gap between supervised and unsupervised methods, when measured by AR. Incorporating these comparisons would further strengthen the contribution of this work.

**Other Strengths And Weaknesses:**

There are small issues with regard to completeness of experiments. For example, the author did not include LVIS results of UnSAM in Table 2, while the UnSAM paper reported such results.

**Questions For Authors:**

N/A

**Relation To Broader Scientific Literature:**

This paper establishes a new state-of-the-art in the field of unsupervised image segmentation by identifying the key limitations of existing approaches such as slot-based ones and clustering-based ones. The proposed two stage solution is well-motivated and novel, and is a clear departure from existing methods. I see this work has meaningful impact to the unsupervised segmentation community.

**Theoretical Claims:**

There are no proofs or theoretical claims.

---

> ### Author Rebuttal · Authors · 2025-04-01
>
> We appreciate the reviewer's thoughtful comments and address all concerns below. An anonymous PDF with figures and tables is available at: <https://github.com/icml5450/icml5450/blob/main/FiguresTables.pdf>
> # Q1: Include UnSAM in Table 2
> A1: We report zero-shot results of UnSAM in the attached ***Table 4*** (will replace Table 2 in the main paper), with two more metrics AR$^{box}$/AR$^{mask}$ as requested. We can see that UnSAM achieves the highest AR$^{box}$ or AR$^{mask}$ scores on all datasets, but its other important metrics are rather low. This is because UnSAM tends to oversegment objects, as also confirmed in the attached ***Table 1***, as well as qualitative results in ***Figure 5*** and ***Figure 6***.
>
>
> # Q2: Add AR$^{box}$ and AR$^{mask}$ in Table 1
> A2: We present the attached ***Table 5*** (will replace Table 1 in the main paper) by adding three more metrics: AR$^{box}$, AR$^{mask}$, and "\# of pred obj.". The AR$^{box}$/AR$^{mask}$ scores are used in the original UnSAM paper to measure the average recall rate without limiting the number of predictions, but the AR$^{box}\_{100}$/AR$^{mask}\_{100}$ scores only consider the top 100 predictions per image and are commonly adopted for object segmentation. In addition, AP scores evaluate the ability to discover more objects with fewer trials (i.e., detections), constituting a balanced view between accuracy and recall. In this regard, existing object segmentation works typically focus on AP scores as they are more informative and less biased.
>
> From the attached ***Table 5***, we can see that UnSAM achieves very high AR$^{box}$/AR$^{mask}$ scores, primarily because it tends to predict an excessive number of objects by grouping granular image segments. This clearly explains its rather low scores on all other critical metrics commonly-used for object segmentation. This is also qualitatively validated in the attached ***Figure 5*** and ***Figure 6***, where UnSAM tends to generate oversegmented patches.
>
> We appreciate UnSAM's effort in effectively reducing the gap between unsupervised and supervised methods in terms of "unlimited detection", mainly measured by AR scores. Nevertheless, our method targets at comprehensive and accurate object discovery, measured by the balanced evaluation protocols for object segmentation.

---

> > ### Comment · Reviewer_CHsG · 2025-04-02
> >
> > Thanks for the response. I keep my recommendation for acceptance. The results and discussions with regard to UnSAM are insightful.

---

> > > ### Author Response · Authors · 2025-04-03
> > >
> > > Dear reviewer CHsG,
> > >
> > > Thank you for dedicating your time and effort to review our paper. We are grateful for your positive feedback and insightful suggestions, which have greatly contributed to the improvement of our manuscript.
> > >
> > > Best,
> > > Authors

---

### Official Review · Reviewer_su1q · 2025-03-14

**Overall Recommendation:** 3

**Summary:**

This paper proposes a multi-object segmentation approach that first trains objectness networks to identify the existence, object center, and object boundary of individual objects, and then use the trained networks to discover objects on images without further training modules. The paper claims that the approach can discover multi-objects more accurately compared to baselines without having access to image annotations.

###  Most of my concerns are addressed. I have updated my rating.

**Claims And Evidence:**

The claims are mostly supported by evidence.

**Essential References Not Discussed:**

N/A

**Experimental Designs Or Analyses:**

Experiments are properly designed to support claims.

**Methods And Evaluation Criteria:**

The proposed methods make sense for the problem.

**Other Comments Or Suggestions:**

For novelty concern, I would like to hear more from other reviewers, and I am open to change my rating.

**Other Strengths And Weaknesses:**

**Strength**

1. The writing is smooth and easy to follow.

2. The paper proposes a pipeline to deal with multi-object segmentation without explicit human annotations.

3. Extensive experiments are conducted to demonstrate the effectiveness of the approach.

**Weekness**

1. The novelty seems somewhat limited considering that the objectness networks have been widely studied in the literature. The proposed approach simply use them for object discovery.

2. The no supervision claim is somewhat overstated. Though the paper emphasizes it is a fully unsupervised approach and does not require human labels, the approach dose involve label obtaining implicitly. First, the use of pretrained models makes the approach less unsupervised and the accuracy of the object representation hinges on the performance of VoteCut; Second, the approach **needs to create a twin negative sample by cropping the largest rectangle on background pixels excluding the tightest object bounding box** as stated in the paper.  Considering these points, is it fair to compare the proposed approach with fully unsupervised approaches?

3. Though the Multi-Object Reasoning Module is training free by directly using the objectness network, it relies on randomly initialization and iterative update of multiple bounding boxes. Does this scale well to images containing a large number of objects? Appendix A.16 studies the iterations of the approach needs. Is there any analysis on segmentation speed?

**Questions For Authors:**

N/A

**Relation To Broader Scientific Literature:**

The object-centric representations discussed in the paper is different from that studied in slot-based methods. The object centric representations learned by slot-based methods cannot only be used for segmentation, but also for generalized composition or generation [1][2][3]. The object centric representations in the proposed paper is mainly for object discovery and segmentation.


[1] Jiang, J. and Ahn, S., 2020. Generative neurosymbolic machines. Advances in Neural Information Processing Systems, 33, pp.12572-12582.

[2] Wang, Y., Liu, L. and Dauwels, J., 2023, July. Slot-vae: Object-centric scene generation with slot attention. In International Conference on Machine Learning (pp. 36020-36035). PMLR.

[3] Wu, Y.F., Lee, M. and Ahn, S., 2024. Neural language of thought models. arXiv preprint arXiv:2402.01203.

**Theoretical Claims:**

No theoretical claims are provided.

---

> ### Author Rebuttal · Authors · 2025-04-01
>
> We appreciate the reviewer's thoughtful comments and address all concerns below. An anonymous PDF with figures and tables is available at: <https://github.com/icml5450/icml5450/blob/main/FiguresTables.pdf>
> # Q1: Representations for segmentation and generation
> A1: This is an interesting point. First, this paper indeed mainly tackles unsupervised multi-object segmentation.
>
> Second, our object-centric representations hold potential for generative tasks. We could train a diffusion model to create object center and boundary fields given various prompts. This approach can be expanded to generate complex multi-object images by first creating individual objects and layouts, allowing for controlled sampling from the learned representations to ensure high-quality boundaries and shapes. However, this is non-trivial and left for future exploration.
> # Q2: Novelty of objectness networks
>
> A2: We will consider an alternative title "Unsupervised Multi-Object Segmentation via Center-Boundary Aware Reasoning", highlighting our core contribution to the challenging task of unsupervised multi-object segmentation and our key technique of center-boundary aware reasoning algorithm.
>
> Regarding objectness networks, as extensively discussed in Section 2 of the main paper, relevant works mainly use the concept of center or boundary in isolation to tackle fully-supervised tasks. In contrast, we extend the concepts of object existence, object center field, and object boundary field in a joint manner to tackle the challenging unsupervised task.
>
> Overall, our proposed two-stage pipeline, object-centric representation learning followed by multi-object reasoning, clearly departs from existing unsupervised methods such as slot-based approaches.
> # Q3: Hinges on VoteCut
> A3: To clarify, our object representation is independent of any specific self-supervised features or grouping strategies. We conduct the following ablation study on four types of pseudo-masks:
> - SelfMask[CVPRW22]: For each image, we employ the strong unsupervised saliency detection model SelfMask to predict a salient region as the pseudo label.
> - MaskCut: For each image, we use the first object discovered by MaskCut as the pseudo label.
> - VoteCut: It's used in our paper.
> - VoteCut+SAM: For each image, a rough mask is generated by VoteCut, and its bounding box is used as a prompt for SAM to predict the final pseudo mask. While this setup yields the best pseudo labels, SAM is a fully supervised model, so this ablation is for reference only.
>
> As shown in the attached ***Table 8***, our method is amenable to all types of rough masks, though their quality affects OCN$\_{disc}$ performance. While SAM scores highest, its improvement over VoteCut is not substantial, as it still relies on bounding box prompts from VoteCut. Importantly, our method does not depend on specific pretrained features, enabling the use of enhanced pretrained models in the future.
> # Q4: Twin negative sample
> A4: Different unsupervised methods leverage self-supervised features and raw images in various ways. In our approach, generating twin negative samples from rough masks is effortless and can be regarded as data augmentation without additional human annotations. Thus, this operation should not be seen as a weakness.
> # Q5: Fairness to fully unsupervised approaches
> A5: Since there is no official definition of what constitutes an unsupervised approach, the fairness criteria for comparison are derived from the extensive body of existing unsupervised methods, notably including recent works such as CutLER, CuVLER, and UnSAM. These methods share a core principle: avoiding the use of human labels while fully utilizing self-supervised features and the derived pseudo labels in various ways.
>
> Our method adheres strictly to this core principle, relying solely on self-supervised features without introducing any additional human annotations. To clarify, all baselines in our paper utilize pretrained self-supervised features. Therefore, it is evidently fair to compare our method with all the unsupervised baselines listed in our paper.
> # Q6: Scale to a large number of objects
> A6: We present a detailed evaluation on COCO* validation dataset based on object count in the attached ***Table 1***. We can see that, as the number of objects per image increases (e.g., $\geq5$ objects), our OCN$\_{disc}$/ OCN consistently outperforms all baselines by growing margins, showing the superiority of our method in dealing with a large number of objects.
> # Q7: Segmentation speed
> A7: Time consumption is detailed in the attached ***Table 3***. Our OCN$\_{disc}$ takes 10 hours to train the objectness network and is slower for Direct Object Discovery. However, our subsequent detector OCN requires only 30 hours to train, benefiting from the high-quality pseudo labels from OCN$\_{disc}$, while baseline detectors take over 60 hours. Ultimately, the inference speed of our OCN matches that of CutLER and CuVLER.

---

> > ### Comment · Reviewer_su1q · 2025-04-02
> >
> > Thanks for the detailed rebuttal. Most of my concerns are addressed. I have updated my rating.

---

> > > ### Author Response · Authors · 2025-04-03
> > >
> > > Dear reviewer su1q,
> > >
> > > We sincerely appreciate your time, efforts, and positive feedback on our paper. Your valuable suggestions and insights have significantly helped us to improve our manuscript.
> > >
> > > Best,
> > > Authors

---

### Official Review · Reviewer_rRHV · 2025-03-16

**Overall Recommendation:** 4

**Summary:**

This paper presents OCN, a new two-stage framework for unsupervised multi-object segmentation in images. The proposed pipeline consists of two stages: the first stage involves learning three levels of object-centric representations—object existence, object center field, and object boundary distance field. In the second stage, a center-boundary aware reasoning algorithm is introduced to iteratively discover multiple objects in single images without relying on neural networks or human annotations. OCN demonstrates superior performance compared to existing unsupervised methods across six benchmark datasets, including COCO, achieving state-of-the-art results in object segmentation, especially in crowded scenes where other methods struggle.

##update after rebuttal: Thanks for the rebuttal. My concern has been addressed mostly. I recommend accept this paper. Great work!

**Claims And Evidence:**

The claim of superior performance compared to existing unsupervised methods is well-supported with quantitative results on 6 benchmark datasets, including COCO. Tables and comparisons are provided.

**Essential References Not Discussed:**

The related work included is sufficient to understand the key contributions.

**Experimental Designs Or Analyses:**

The design is robust, and it is particularly important to evaluate whether the method performs exceptionally well in scenarios involving crowd images (i.e., cases with multiple objects), as demonstrated in Table 5 of Appendix A.8.

**Methods And Evaluation Criteria:**

Yes, the proposed methods and evaluation criteria in the paper make sense for the problem of unsupervised multi-object segmentation in single images

**Other Comments Or Suggestions:**

The paper could provide more discussion of failure cases and limitations, which would give a more balanced view of the capabilities.

**Other Strengths And Weaknesses:**

Strengths:

1. The paper provides a detailed explanation of the method's motivation and description. The experiments are comprehensive, including comparisons with state-of-the-art methods across diverse benchmarks and an ablation study.

2. The method is both sound and intuitive. Using three levels of objectness to feed into the object reasoning network makes sense, and leveraging DINO's self-supervised features allows it to perform well in crowded scenarios.

Weaknesses:

1. The three levels of object priors are heavily reliant on the pretrained features. This dependence could introduce biases from the training dataset, potentially limiting the model’s generalization capabilities.

**Questions For Authors:**

1. Does the method heavily rely on VoteCut when training the Objectness Network? Is there an ablation study to replace the pseudo-masks generated by other methods?

2. In Appendix A.16, the paper mentions that the average number of iterations in the object reasoning module is typically 10. Could this be considered time-consuming? Additionally, what is the average throughput of this module?

**Relation To Broader Scientific Literature:**

I think the the finding in this paper can be applied in the video domain, too.

**Theoretical Claims:**

Not applicable.

---

> ### Author Rebuttal · Authors · 2025-04-01
>
> We appreciate the reviewer's thoughtful comments and address all concerns below. An anonymous PDF with figures and tables is available at: <https://github.com/icml5450/icml5450/blob/main/FiguresTables.pdf>
> # Q1: Applied on videos
> A1: We agree with the reviewer and conduct the following experiments on YouTubeVIS-2021 dataset to verify the effectiveness of OCN for unsupervised video object segmentation.
> - Dataset: The YouTubeVIS-2021 dataset consists of 2,985 training videos and 421 val videos whose labels are held for competitions. Thus, we split original training videos into two subsets: YouTubeVIS-2021 Train\# (2,795 videos) and YouTubeVIS-2021 Val\# (200 videos).
> - Baselines: 1) We compare with CutLER's extension to video domain: VideoCutLER(CVPR24). 2) We also adapt CuVLER to video domain: VideoCuVLER. 3) We adapt OCN to video domain: VideoOCN.
> - Experiments: We follow VideoCutLER: Step-1: generate pseudo labels for unlabeled images. Step-2: generate synthetic videos with images and pseudo labels from Step-1. Step-3: train a Mask2Former model for video segmentation on synthetic videos and labels from Step-2.
> - Results: As shown in the attached ***Table 7***, the baselines VideoCutLER and VideoCuVLER are trained on 2 types of training sets, ImageNet and YouTubeVIS-2021 Train\#. VideoOCN consistently outperforms all baselines on most metrics without extensively tuning hyperparameters due to limited time. Qualitative results for video segmentation can be found in the attached ***Figure 7***.
> # Q2: Pretrained features and generalization
> A2: This is an insightful point. Like almost all pretrained models, the learned features are always depending on training datasets, and can hardly generalize to extremely different domains due to the fundamental data-driven learning principle. As shown in the attached ***Table 4***, our OCN (trained on natural images) demonstrates excellent zero-shot performance on unseen datasets with diverse types of natural images. Nevertheless, for datasets with significant domain gaps (e.g., medical images), our learned object priors from natural images may not achieve comparable results as expected. To enhance generalization, one potential solution could be to increase the diversity of training datasets. However, this is a non-trivial task and is left for future exploration.
> # Q3: Failure cases and limitations
> A3: We present failure cases in the attached ***Figure 8*** and discuss limitations as follows.
>
> 1. The Direct Object Discovery of our OCN$\_{disc}$ takes time. It could be possible to leverage reinforcement learning techniques to learn an efficient policy network to discover objects.
>
> 2. Our method struggles to separate overlapping objects with similar textures, as shown in the attached ***Figure 8***. Adding additional language priors may help alleviate this issue.
> # Q4: Ablation study on pseudo-masks
> A4: We conduct the following ablation study on four types of pseudo-masks:
> - SelfMask[CVPRW22]: For each image, we employ the strong unsupervised saliency detection model SelfMask to predict a salient region as the pseudo label.
> - MaskCut: For each image, we use the first object discovered by MaskCut as the pseudo label.
> - VoteCut: It's used in our paper.
> - VoteCut+SAM: For each image, a rough mask is generated by VoteCut, and its bounding box is used as a prompt for SAM to predict the final pseudo mask. While this setup yields the best pseudo labels, SAM is a fully supervised model, so this ablation is for reference only.
>
> As shown in the attached ***Table 8***, our method is amenable to all types of rough masks, though their quality affects OCN$\_{disc}$ performance. While SAM scores highest, its improvement over VoteCut is not substantial, as it still relies on bounding box prompts from VoteCut. Importantly, our method does not depend on specific pretrained features, enabling the use of enhanced pretrained models in the future.
> # Q5: Time consumption and throughput
> A5: We present the time consumption in the attached ***Table 3***. Our OCN$\_{disc}$ takes 10 hours to train the objectness network and is slower for Direct Object Discovery. However, our subsequent detector OCN requires only 30 hours to train, benefiting from the high-quality pseudo labels from OCN$_{disc}$, while baseline detectors take over 60 hours. Ultimately, the inference speed of our OCN matches that of CutLER and CuVLER.
>
> Regarding the throughput, for each image on average, the number of initial proposals is 1122.7, whereas the number of predicted objects from OCN$_{disc}$ is 8.9. Most initial proposals have low existence scores and are discarded at the first iteration. The Non-Maximum Suppression (NMS) will also remove redundant proposals.

---

### Official Review · Reviewer_QK7x · 2025-03-17

**Overall Recommendation:** 3

**Summary:**

The paper proposes OCN, which improves unsupervised multi-object discovery by introducing three objectness scores to measure existence, centers, and boundaries, along with a reasoning module to distinguish objects. The model is trained by bootstrapping rough masks from DINOv2 and refined through distillation with inductive biases, leading to superior performance over CutLER and CuVLER.

**Claims And Evidence:**

The paper does not focus on "object-centric representation" as it only learns object segments, unlike SlotAttention, which learns embeddings for each segment. The title and terminology should be corrected to reflect its focus on "unsupervised multi-object segmentation."

The main contribution is improving unsupervised segmentation in scenes with many objects, but this is not thoroughly analyzed:
- The tables present only average performance per dataset. Showing performance based on object count would be more informative. While the paper uses COCO*, a multi-object extension of COCO, the analysis is insufficient.
- The figures display only object centers and boundaries for a single object, including those in Appendix A.13. More qualitative results demonstrating objectness in multi-object segmentation would be beneficial.

**Essential References Not Discussed:**

Well-cited, as far as I know.

**Experimental Designs Or Analyses:**

Mentioned above.

**Methods And Evaluation Criteria:**

1. The paper aims for fine-grained object discrimination through objectness measurement, but the results are unconvincing due to the lack of performance analysis by object count and qualitative results on multi-object images, as mentioned above.
2. The model relies on rough masks for supervision, which may introduce errors if they fail to distinguish objects. It would help to show that while the original masks suffer from merging issues, refinement through objectness improves segmentation both qualitatively and quantitatively.
3. The paper introduces multiple complex modules beyond prior work. It should compare training and inference time, not just list trainable modules in Table 1.
4. While CutLER and CuVLER suffer from undersegmentation, UnSAM does not. Why does Table 2 omit a comparison with UnSAM?
5. Why is UnSAM’s performance in Table 1 so low? Can the authors justify this? Also, why reimplement its results instead of comparing directly with the benchmarks reported in the UnSAM paper?

**Other Comments Or Suggestions:**

Mentioned above.

**Other Strengths And Weaknesses:**

I appreciate that the authors have updated the paper rather than simply resubmitting a rejected version, addressing prior reviews. Here are my thoughts on additional strengths and weaknesses after reading A.17.

Technical contribution (strength):
- While concerns about technical novelty are valid, the paper makes a reasonable contribution by demonstrating how combining objectness components improves multi-object discovery benchmarks.
- Prior work is well surveyed and properly discussed in Sec 3.2.

Presentation (weakness):
- The presentation could be further refined for clarity and readability.
- Figure 1 lacks clarity and should convey a clear message without relying on the text.
- The ablation study in Table 3 is difficult to read. Instead of describing variants 1–8 in the text, clarify them directly in the table using checkmarks or dashes.
- Visualizations in Figures 11–17 are strong. Consider curating some for the main paper to provide additional insights beyond numerical results. Adding more baselines, especially UnSAM, would further illustrate how OCN outperforms it qualitatively.

**Questions For Authors:**

Questions 1-5 in "Methods And Evaluation Criteria."

Visual comparison between OCN and UnSAM (extension of Figures 11–17).

**Relation To Broader Scientific Literature:**

Object discrimination is a fundamental problem in computer vision with applications across various visual tasks, including scientific problems like cell segmentation.

**Theoretical Claims:**

N/A

---

> ### Author Rebuttal · Authors · 2025-04-01
>
> We appreciate the reviewer's thoughtful comments and address all concerns below. An anonymous PDF with figures and tables is available at: <https://github.com/icml5450/icml5450/blob/main/FiguresTables.pdf>
> # Q1: Title and terminology
> A1: Thanks for this advice. We will consider an alternative title "Unsupervised Multi-Object Segmentation via Center-Boundary Aware Reasoning". In addition, we will ensure that the relevant terminology throughout the paper is updated accordingly.
> # Q2: Table and Figure based on object count
> A2: We present a detailed evaluation on COCO* validation dataset based on object count in the attached ***Table 1*** with two more metrics AR$^{box}$/AR$^{mask}$ as requested by reviewer **CHsG**. We can see that, when the number of objects in each image is rather small (e.g., [0 - 4]), the results of top-performing baselines VoteCut/CuVLER are comparable to our method, all yielding high scores. However, as the number of objects per image increases (e.g., $\geq5$ objects), our OCN$_{disc}$/ OCN consistently outperforms all baselines by growing margins, demonstrating the superiority of our method in dealing with challenging crowded images.
>
> Notably, UnSAM achieves high AR$^{box}$/AR$^{mask}$ scores (used in the original UnSAM paper to measure the average recall rate without limiting the number of predictions), but its AR$^{box}\_{100}$/AR$^{mask}\_{100}$ scores (only considers the top 100 predictions per image and commonly adopted for object segmentation) are clearly lower. This is because UnSAM focuses on excessively partitioning images by clustering granular segments, which sacrifices the accuracy of object discovery, but tends to oversegment objects. This is also qualitatively validated in attached ***Figure 5*** and ***Figure 6***.
>
> We present more qualitative results in the attached  ***Figure 1*** for multi-object reasoning.
> # Q3: Refinement through objectness
> A3: This is an insightful point. We compare VoteCut and our OCN$\_{disc}$ on COCO train2017 and ImageNet val splits. As shown in the attached ***Table 2***, our OCN$\_{disc}$ is on par with VoteCut on ImageNet val, validating that our OCN$\_{disc}$ indeed learns valid objectness from rough masks generated by VoteCut on the train split of ImageNet. Since most images of ImageNet have a single object, it is expected that our OCN$\_{disc}$ performs similarly to the pseudo label generator VoteCut. However, on the challenging COCO train2017, our OCN$\_{disc}$ clearly outperforms VoteCut, validating that the learned (refined) objectness by our OCN$\_{disc}$ can better deal with undersegmentation on multi-object images, whereas VoteCut cannot.
>
> As shown in the attached ***Figure 2***, rough masks from VoteCut  on both COCO train2017 and ImageNet val are prone to undersegmentation, while OCN$\_{disc}$ shows a stronger ability to distinguish multiple objects.
> # Q4: Training and inference time
> A4: We report the training and inference time in the attached ***Table 3***. Our OCN$\_{disc}$ takes 10 hours to train the objectness network and is slower for Direct Object Discovery. However, our subsequent detector OCN requires only 30 hours to train, benefiting from the high-quality pseudo labels from OCN$_{disc}$, while baseline detectors take over 60 hours. Ultimately, the inference speed of our OCN matches that of CutLER and CuVLER.
> # Q5: Add unSAM to Table 2 of the main paper
> A5: We report zero-shot results of UnSAM in the attached ***Table 4*** (will replace Table 2 in the main paper), with two more metrics AR$^{box}$/AR$^{mask}$ as requested by reviewer **CHsG**. We can see that UnSAM achieves the highest AR$^{box}$ or AR$^{mask}$ scores on all datasets, but its other important metrics are rather low. This is because UnSAM tends to oversegment objects, as also confirmed in the attached ***Table 1***.
> # Q6: unSAM in Table 1 of the main paper and its reproduction
> A6: To clarify, all results of UnSAM in our paper are based on its official checkpoints and code. We will rephrase sentences.
>
> In the attached ***Table 5*** (will replace Table 1 in the main paper), we add three more metrics: AR$^{box}$, AR$^{mask}$, and "\# of pred obj.". Again, we can see that UnSAM achieves very high AR$^{box}$/AR$^{mask}$ scores, primarily because it tends to predict an excessive number of objects. This clearly explains its rather low scores on all other critical metrics commonly-used for object segmentation.
> # Q7: Figure 1 improvement
> A7: We present an updated version in the attached ***Figure 3*** which will replace Figure 1 of the main paper.
> # Q8: Table 3 improvement
> A8: We present an updated version in the attached ***Table 6*** which will replace Table 3 of the main paper.
> # Q9:  Visualizations in Figures 11–17
> A9: We present new visualizations in the attached ***Figure 4/5/6*** by re-organizing existing materials and adding results of UnSAM on both COCO* validation and zero-shot datasets. We will include them in the main paper for better illustration.

---

### Decision · Program_Chairs · 2025-05-01

**Decision:**

Accept (poster)

**Comment:**

This paper presents a two-stage unsupervised framework that learns hierarchical object-centric representations and performs network-free multi-object reasoning to achieve state-of-the-art segmentation on complex real-world images without using labels.

It has received 4 reviews, with 1x accept and 3x weak accepts.  Authors provided extensive rebuttals with additional experimental results on unSAM and videos etc, convincing reviewers on an acceptance consensus.  The network-free reasoning aspect using object existence, object centers, and object boundaries is interesting and seems effective.